# Convolutional neural networks explain tuning properties of anterior, but not middle, face-processing areas in macaque inferotemporal cortex

Rajani Raman [1]✉ & Haruo Hosoya [1]✉

Recent computational studies have emphasized layer-wise quantitative similarity between convolutional neural networks (CNNs) and the primate visual ventral stream. However, whether such similarity holds for the face-selective areas, a subsystem of the higher visual cortex, is not clear. Here, we extensively investigate whether CNNs exhibit tuning properties as previously observed in different macaque face areas. While simulating four past experiments on a variety of CNN models, we sought for the model layer that quantitatively matches the multiple tuning properties of each face area. Our results show that higher model layers explain reasonably well the properties of anterior areas, while no layer simultaneously explains the properties of middle areas, consistently across the model variation. Thus, some similarity may exist between CNNs and the primate face-processing system in the near-goal representation, but much less clearly in the intermediate stages, thus requiring alternative modeling such as non-layer-wise correspondence or different computational principles.

[1] ATR Cognitive Mechanisms Laboratories, Hikaridai 2-2-2, Seika-cho, Kyoto 619-0288, Japan. ✉email: rajaniraman@ymail.com; hosoya@atr.jp

Recently, the neuroscience community has witnessed the rise of the deep convolution neural network (CNN)[1], a family of feedforward artificial neural networks, in computational modeling of the primate visual system. CNN models trained for behavioral goals have exhibited remarkable similarity to ventral visual areas in terms of stimulus-response relationship despite that the network itself was not directly optimized to fit neural data. For example, CNN models trained for image classification were highly predictive of single-site and population-level neural responses in the inferotemporal (IT) cortex[2,3]. Parallels along the hierarchy have been discovered between layers of CNN and the intermediate[3–5] or lower visual areas[6]; analogous parallels have also been reported in a decoding study of fMRI signals[7]. Such predictive CNN model has further been used to generate "optimal" stimuli for model validation[8]. A natural question arises here: if CNN explains overall responses in IT, then does it also explain responses in a subsystem of IT?

Among various subsystems of IT[9–11], the most well-studied is the macaque face-processing system[12,13]. This subsystem forms a network consisting of multiple face-selective patches with anatomically tight inter-connections[14]. The network putatively has a functional hierarchy from the middle to the anterior patch areas with a progressive increase of selectivity to facial identities and invariance in viewing angles[15]. For each patch area, a number of tuning properties to specific facial features have been documented in a clear and detailed manner[15–18]. Given these experimental facts, the macaque face processing system emerges as an ideal testbed to examine our question regarding the generality of CNN as a model of higher visual processing.

Thus, in this study, we have asked whether CNN explains previously reported tuning properties of face neurons in macaque IT (Fig. 1). More specifically, we explored a variety of CNN models that were trained for classification with different architecture and dataset settings. We incorporated four major physiological experiments that had been conducted on the middle lateral (ML), anterior lateral (AL), and anterior medial (AM) patches: (1) view-identity tuning in ML, AL, and AM[15], (2) shape-appearance tuning in ML and AM[18], (3) facial geometry tuning in ML[16], and (4) contrast polarity tuning in ML[17]. While simulating these experiments on each model, we attempted to make a correspondence between the model layers and the macaque face patches by matching the population-level tuning properties. In particular, by exploiting the available multiple experimental results on the same face patch, we aimed at performing a strong plausibility test on each model layer. Our results show that, for most of the explored CNN models, higher layers give a reasonably good match with the multiple tuning properties of AM, in particular, tuning related to invariance and appearance.

However, no single layer matches well the multiple properties of ML simultaneously: the best-matching layer for each different property of ML is either a lower, intermediate, or higher one. Thus, although the near-goal representation in face processing may be somewhat similar between CNN and macaque, such similarity may be much weaker for the intermediate representation. Although a possibility for a more complicated, non-layerwise correspondence still remains, our result motivates us to consider alternative approaches to model the computation in the primate face-processing system.

## Results

To investigate whether CNN can explain known tuning properties of the macaque face-processing network, we started with a representative CNN model optimized for classification of face images. Our model adopted an architecture similar to AlexNet[19], following recent studies relating CNNs with the ventral stream[2,4,7]. The AlexNet architecture has seven layers in total. The first five "convolutional" layers perform multi-channeled local linear filters that are replicated across the visual field. Repetition of such layers progressively increase the size of visual receptive fields, mimicking the general structure of the visual cortex. Then, two "fully connected" layers follow and cover the entire visual field. (The network ends with a special layer representing the class, which is ignored throughout our analysis.) We trained our CNN model on a large number of natural face images for classifying facial identities (using the VGG-Face dataset with data augmentation for size variation; see "Methods"). From here on, we refer to this network as "AlexNet-Face."

For our AlexNet-Face model thus constructed, we first identified a population of face-selective units in each layer ("Methods"); we call face-selective population simply "population" and face-selective unit simply "unit" from here on. We then ran the protocols (stimulus set and data analysis) of previous four monkey experiments[15–18] on our model and thereby investigated whether each model layer replicated similar population-level tuning properties to the corresponding published experimental data (Fig. 1). Note that, in this approach, we need no raw experimental data.

**View-identity tuning**. In the first study that we consider[15], it has been reported that different macaque face patch areas (ML, AL, and AM) have different joint tuning properties to the facial view and identity. Accordingly, we incorporated the same set of face images as used experimentally, which consisted of 25 identities and 8 views (frontal, left, and right half-profiles, left and right profiles, up, down, and back). For each layer of our CNN model, we first recorded the responses of all model units to those images.

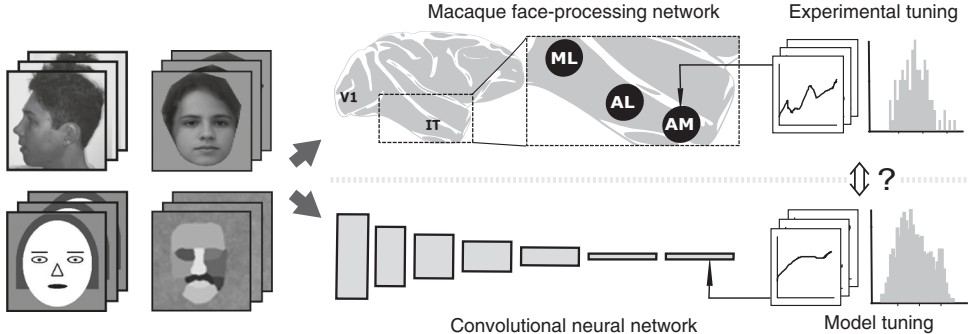

**Fig. 1 Schema of our investigation to compare the macaque face-processing network and a CNN model.** We simulate previous four experiments (left image sets) on a CNN model (bottom middle) to identify tuning properties (bottom right). We quantitatively compare the tuning properties between each macaque face patch (from the past experiment) and each CNN layer (from the present simulation) to find out their correspondence.

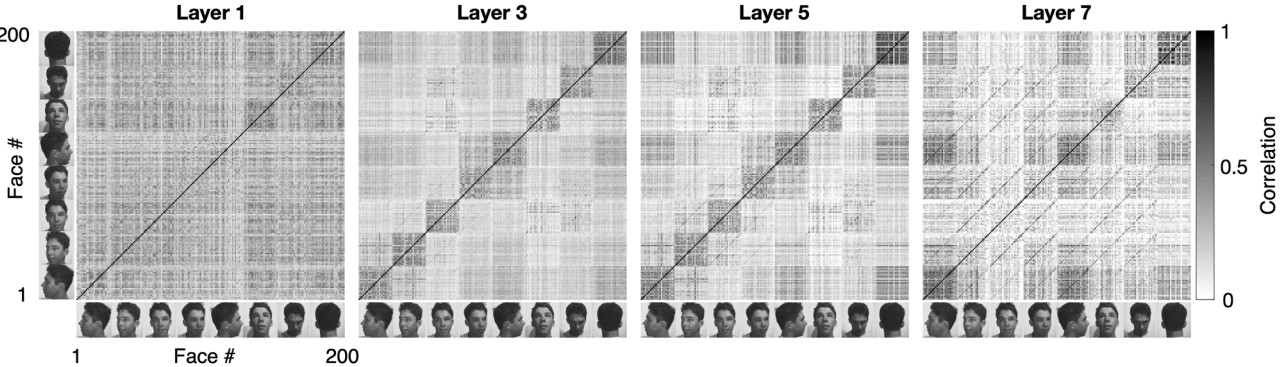

**Fig. 2 View-identity tuning.** Each plot shows the population response similarity matrix for each layer. The pixel values of the matrix indicate the pairwise correlation coefficients (legend) for the population responses to face images. The elements of the matrix are grouped according to the view (indicated by the images along the axes) with the same order of identities in each group.

We then calculated the correlation between the population responses to each pair of face images and constructed a population response similarity matrix (RSM) of such correlations for all pairs ("Methods").

Figure 2 shows RSMs for layers 1, 3, 5, and 7, where the face numbers are grouped according to the view (forming sub-matrices of size $25 \times 25$) and sorted by identity in the same way within each group. (For succinctness, we present, here and hereafter, comparison results only for the odd-numbered layers since the remaining layers generally give more or less interpolated results of the presented layers.) In the intermediate layers (layers 3–5), the RSMs had strong block-diagonal patterns, indicating that most units had selectivity to a specific view, similarly to the corresponding RSM for ML[15] (correlation between the RSMs: 0.56, layer 3; 0.55, layer 5; see "Summary of Correspondence" for more on the quantification). In the top layer (layer 7), such block-diagonal structure disappeared, but para-diagonal lines instead became prominent, indicating that most units had selectivity to facial identity with some degree of view invariance, similarly to AM[15] (RSM correlation: 0.53). However, at a closer look, the response similarities between the profile and frontal views were weaker than those between the half-profile and frontal views, which is likely due to our training image set including fewer images in profile views than half-profile views (see relevant results in Supplementary Fig. 1). In addition, a mirror-symmetric pattern in view can be seen from layers 3 to 7, somewhat similarly to AL[15] (RSM correlation: 0.51, layers 3, 5; 0.36, layer 7), though such symmetry was apparent only between the left and right profile views, not the half-profile views. In sum, the intermediate-to-top layers gradually shifted from view-specific to view-invariant and identity-selective, which is reasonably consistent with the idea of functional hierarchy in the macaque face patches[15].

The same experimental study has also reported invariance property in stimulus size[15]. Thus, we recorded the model unit responses to a set of face and non-face object images of various sizes (illustrated in Fig. 3a). Then, for each layer and for each image size, we calculated the response, $\bar{R}_{face}^{size}$, to face images averaged over the population and the stimulus set; similarly, we calculated the average response, $\bar{R}_{object}^{size}$, to object images. We quantified the degree of size invariance by how much robustly the population-level selectivity to faces over objects retained for different sizes: size-invariance index (SII) is defined as the minimal fraction of image sizes at which the average response to faces is reasonably larger than that to objects ($\bar{R}_{face}^{size} > 1.4 \bar{R}_{object}^{size}$); thus, a lower SI-index indicates a stronger size invariance ("Methods").

Not surprisingly, size invariance in our CNN model strengthened along with its depth (Fig. 3b). In particular, the top layer

(layer 7) had the strongest size invariance (SII: 1/4), where the average response to faces (red) was always larger than to objects (blue) for all tested sizes. The top layer also quantitatively came closest, of all layers, to the face patches (ML/AL/AM), which all give SII around 1/8[15] (see "Summary of Correspondence" for more on the quantification). (Supplementary Fig. 2 shows additional results on size invariance as well as position invariance.)

**Shape-appearance tuning.** In the second experimental study[18], coding of facial shapes and appearances in the macaque face patches (ML and AM) has been investigated. Similarly to their method, we constructed a face space based on the active appearance model[20]. The face space was described by 50-dimensional feature vectors, consisting of 25 shape and 25 appearance dimensions, defined as follows. Using a set of natural frontal face images with coordinates annotated on pre-defined facial landmarks, the shape dimensions were the first 25 principal components (PC) of the landmark coordinates and the appearance dimensions were the first 25 PCs of the original face images that were morphed so that the landmarks matched to their mean coordinates ("Methods"); Fig. 4a illustrates the first shape and first appearance dimensions. We then randomly sampled a set of face images from this space and used it for all the subsequent analyses.

Following the experimental study[18], we examined whether and how much each model unit preferred shape or appearance. We first recorded the responses to the face images and estimated the 50-dimensional vector of spike-triggered average (STA), i.e., the average of feature vectors of the face images weighted by the responses. We then computed the shape preference index (SPI), $(S - A)/(S + A)$, where $S$ is the vector length of the shape dimensions and $A$ is the vector length of the appearance dimensions of the STA; a unit is considered to prefer shape when SPI is positive and prefer appearance when SPI is negative ("Methods").

Figure 4b shows the distribution of SPIs for each layer (blue). In the intermediate-to-higher layers (layers 5–7), most units had appearance preference (mean SPI: −0.15, layer 5; −0.27, layer 7; see "Summary of Correspondence"), with a distribution much closer to the corresponding experimental data on AM (pink; mean SPI: −0.19) than ML (gray; mean SPI: 0.29)[18]. However, the lower layers (layers 1–3) broadly mixed both shape-preferring and appearance-preferring units (mean SPI: 0.12, layer 1; −0.04, layer 3), thus showing an intermediate property between AM and ML, but somewhat closer to ML in layer 1 and to AM in layer 3 (both significantly deviating from the midpoint of ML and AM;

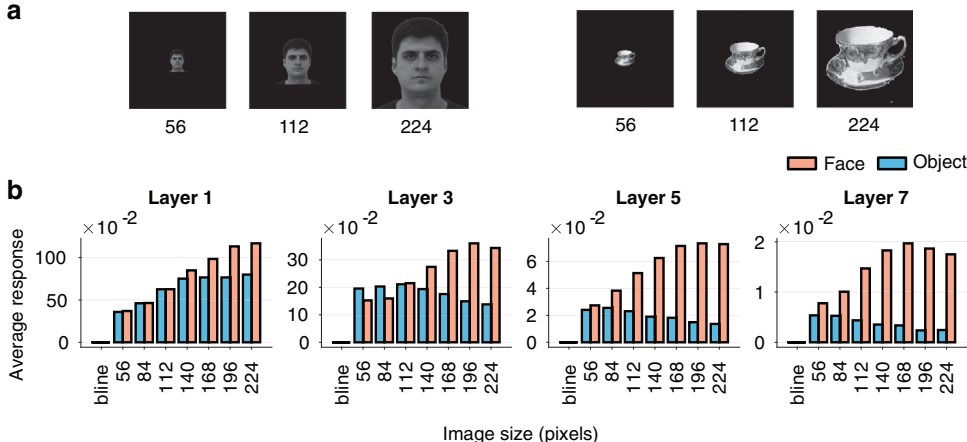

**Fig. 3 Size invariance. a** Examples of face and non-face object stimuli of various sizes (the numbers in pixels beneath). **b** The average responses to the face images (red) and to the object images (blue) for each image size (x-axis) in each model layer. The "bline" stands for the average baseline response to the blank image.

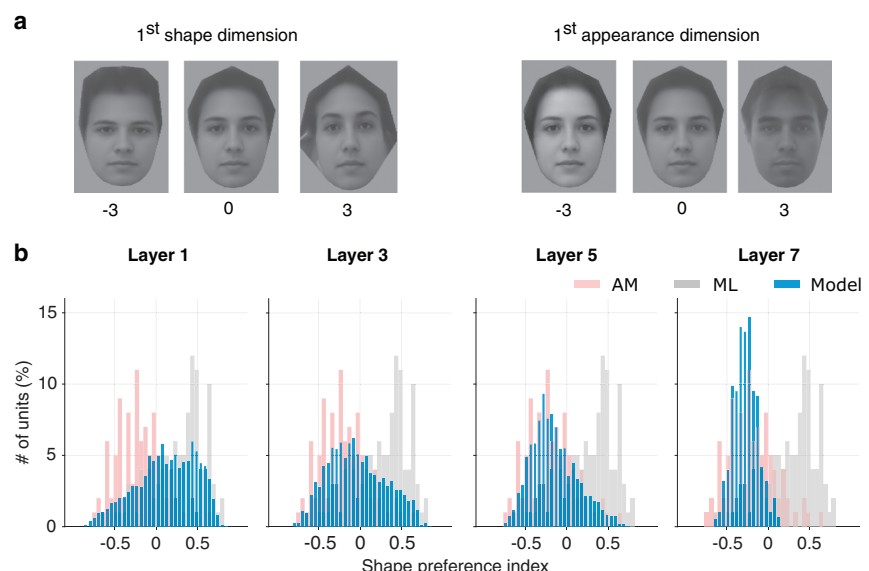

**Fig. 4 Shape-appearance tuning. a** Illustration of the first shape and first appearance dimensions for frontal faces. It shows how varying these dimensions changes the image. The shown images correspond to the feature vectors with all zero except the indicated dimension is set to −3, 0, or 3. **b** The distribution of shape-preference indices for each model layer (blue), in comparison to the corresponding distributions for ML (gray) and AM (pink) estimated using Fig. 1e of the experimental study[18].

see "Summary of Correspondence" for the statistical analysis). We also examined shape-appearance preference by counting the number of significantly tuned units to each feature dimension; the tendency was similar (Supplementary Fig. 3). Furthermore, we investigated how much information on the face space was contained in each layer by decoding feature vectors from population responses; the top layer exhibited a better match with AM than ML, but lower-to-intermediate layers showed large discrepancies from both AM and ML (Supplementary Fig. 4). In sum, in terms of shape-appearance tuning, the CNN model exhibited prominently AM-like properties but less clearly ML-like properties. (See also Supplementary Fig. 5 for additional results on ramp-shape tuning properties).

The same experimental study also investigated view tolerance in the face space representation in AM[18]. Accordingly, we first constructed the feature representation for profile faces in a way compatible with frontal faces. Namely, we built another shape-

appearance face space from left profile face images, similarly to frontal faces, and established a mapping between the frontal and profile face spaces via linear regression; we hereafter always used feature vectors for profile faces that were mapped to the frontal face space, which allowed us to use the same feature vectors for both views of the same identity ("Methods").

Figure 5a illustrates the first shape and appearance dimensions for profile faces; note their compatibility with the frontal faces in Fig. 4a. For each unit, we estimated the STA from profile faces, similarly to frontal faces, and calculated the correlation between the frontal and the profile STAs at each feature dimension across all units in each layer (Fig. 5b, red and blue). As a result, the STA correlations at the first half of the appearance dimensions (blue) increased from lower to higher layers (mean STA correlation: 0.078, layer 1; 0.22, layer 7; see "Summary of Correspondence"), indicating a gradual progression of view tolerance along with the depth. In particular, the correlations in the top layer (layer 7)

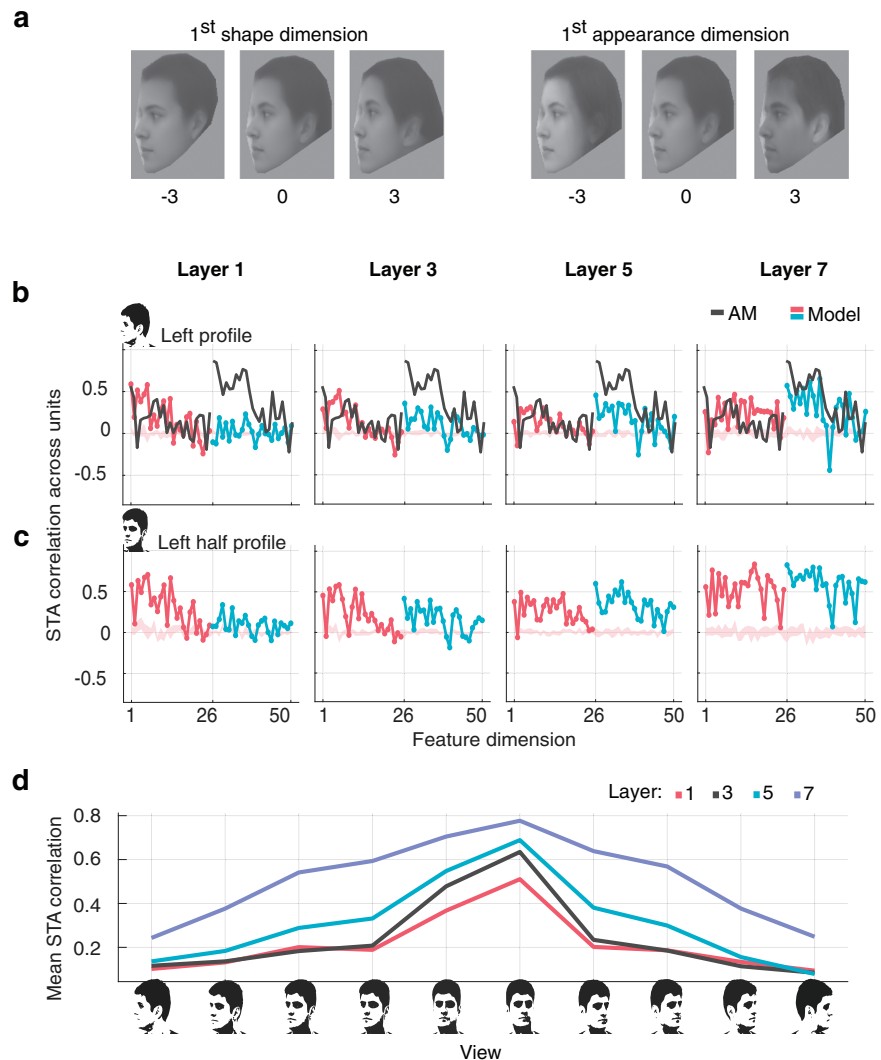

**Fig. 5 View tolerance. a** Illustration of the first shape and first appearance dimensions for profile faces. The images correspond to the feature vectors with all zero except the indicated dimension is set to −3, 0, or 3. **b** The correlation between the frontal and the profile STAs across units for each dimension (x-axis; 1–25: shape, 26–50: appearance). Each plot compares the results from a model layer (red and blue) and AM (black), the latter replotted from Fig. 6d of the experimental study[18]. The shaded region indicates the 99% confidence interval of randomly shuffled data from the model. **c** Analogous result for the left half-profile view. **d** The mean STA correlation (averaged over the feature dimensions) for each non-frontal view (x-axis) against frontal view, for each layer (color; see legend).

came closest, of all layers, to the corresponding data on AM (black; mean STA correlation: 0.31), which is consistent with the view-identity tuning in the top layer (Fig. 2). We also conducted a decoding analysis using a mixed set of frontal and profile faces; only the top layer showed similar decoding performance between both views (Supplementary Fig. 6), consistently with AM[18].

To gain further insight, we repeated the above analysis for all available 10 non-frontal views including half-profile views. The result for a left half-profile view (Fig. 5c) shows a similar tendency to the left full-profile, but the overall STA correlations were slightly higher, indicating stronger tolerance for the half-profile view than the full-profile view. This result is again consistent with Fig. 2, in which the frontal view is more strongly correlated with the half-profile view than the full-profile view. Finally, Fig. 5d plots the mean STA correlation between the frontal and the non-frontal STAs (averaged over the feature dimensions) for all non-frontal views and for all layers. The mean STA correlation increased, thus view-tolerance became stronger, in a higher layer and in a view closer to the frontal view.

**Facial geometry tuning**. In the third experimental study[16], tuning of ML neurons to local and global features in cartoon face stimuli has been documented. Thus, we incorporated their stimulus design of cartoon face images parametrized by 19 different facial features (Fig. 6a). We randomly generated a set of such cartoon face images. From the responses of each unit to those images, we estimated a tuning curve for each feature parameter and determined its statistical significance ("Methods").

Figure 6b (blue) shows how many features each unit was significantly tuned to (features-per-unit (FPU)) in each model layer. Analogously, Fig. 6c (blue) shows how many units in each layer were significantly tuned to each feature (units-per-feature (UPF)). In both plots, upper layers showed very different distributions from ML (gray; cosine similarity in FPU: 0.50, layer 5; 0.49, layer 7; in UPF: 0.59, layer 7; not significantly larger than random cases; see "Summary of Correspondence" for the statistical analysis), accommodating more units tuned to larger numbers of features and to more remaining features absent in ML. Curiously, lower layers gave results somewhat

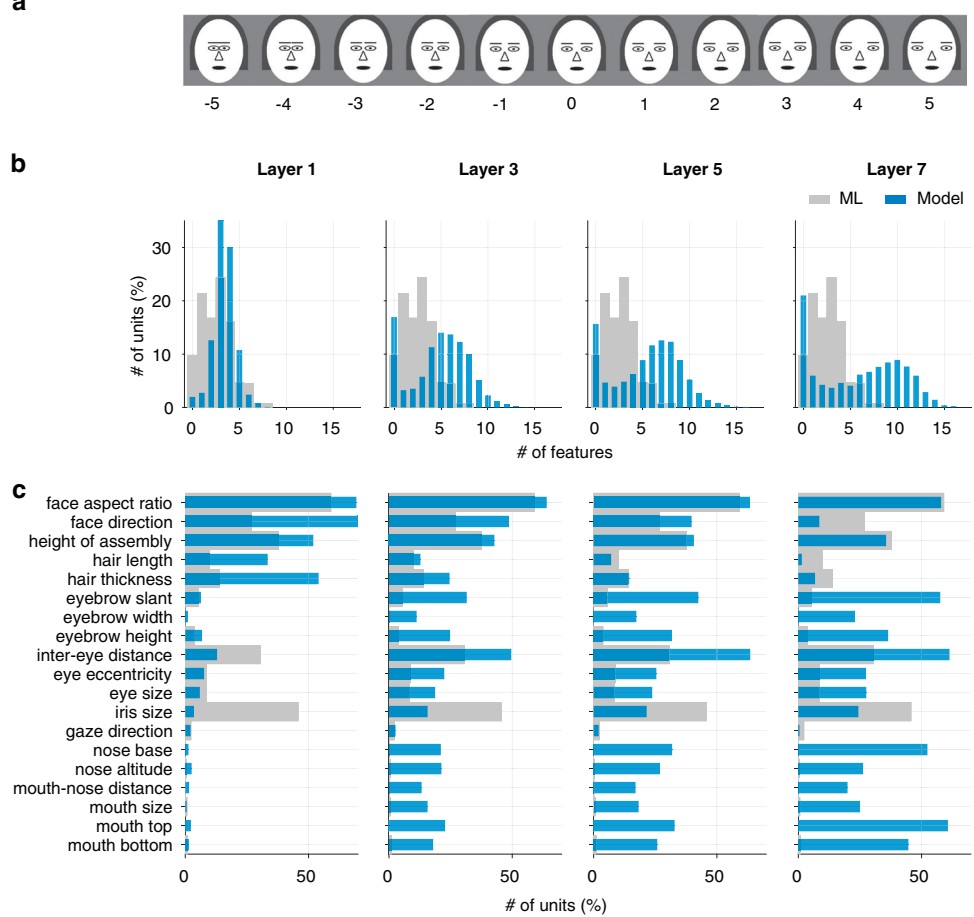

**Fig. 6 Facial geometry tuning. a** Examples of cartoon face images varying a feature parameter (inter-eye distance in this case). Generally, each of 19 feature parameters ranges from −5 to +5, where ±5 corresponds to the extreme features and 0 corresponds to the mean features. **b** The distribution of the number of features that each unit is significantly tuned to. **c** The distribution of the number of units significantly tuned to each feature. Each plot compares the result from a model layer (blue) with that from ML (gray) replotted from Fig. 3 of the experimental study[16].

closer to ML (cosine similarity in FPU: 0.82, layer 1; in UPF: 0.78, layer 1; 0.82, layer 3; 0.76, layer 5; significantly larger than random cases): most units were tuned to a small number of features, which were mostly geometrically larger features (i.e., face aspect ratio, face direction, feature assembly height, and inter-eye distance) rather than smaller features (related to mouth and nose). In addition, in all layers, a majority of units had ramp-shaped tuning curves with peaks and troughs at the extreme values (Supplementary Fig. 7), similarly to ML[16] (see "Discussions").

**Contrast polarity tuning**. The last experimental study[17] has reported that ML units had preference for contrast polarities between face parts in mosaic-like cartoon face stimuli[17]. Thus, we again used their stimulus design of cartoon face images, which consisted of 11 distinct face parts that were each assigned a unique intensity value varying from dark to light (Fig. 7a). We randomly generated a set of such face images and analyzed the responses of each unit to those images for identifying its contrast polarity tuning. That is, for each pair of face parts, A and B, out of 55 pairs in total, we determined whether the unit prefers part A lighter than part B (part A > part B) or the opposite (part A < part B); see "Methods".

As summarized in Fig. 7b (blue and red), in all layers, most of the units had preferences for contrast polarities mainly related to the forehead, the largest geometrical area in the mosaic-like face.

This result is inconsistent with the experimental finding in ML[17] (gray; cosine similarity: <0.51, layers 1, 3, 5, 7; not significantly larger than random cases; see "Summary of Correspondence" for the statistical analysis), where most neurons were tuned to eye- or nose-related contrast polarities and the polarity directions were consistent across the neurons.

**Summary of correspondence**. To see more clearly which model layer corresponds to each macaque face patch, we next quantify the similarities of tuning properties. Figure 8 summarizes the results for our AlexNet-Face model. For each tuning property, we use a different metric to quantify similarity between the results from the model and the experiment. For view-identity tuning (Fig. 8a), we use the correlation between the response similarity matrices from each layer (Fig. 2) and each face patch[15]. For size invariance (Fig. 8b), we compare the size invariance indices for each layer (Fig. 3b) and each face patch[15]. For shape-appearance preference (Fig. 8c) or view tolerance (Fig. 8d), we compare the averages of shape-preference indices (Fig. 4b) or mean STA-correlations (Fig. 5b) from each layer and the corresponding experimental data[18]. For facial geometry tuning (Fig. 8e) and contrast polarity tuning (Fig. 8f), we use the cosine similarity between the distributions from each layer (Figs. 6 and 7, respectively) and each face patch[16,17]. For most of these, we introduce statistical criteria to test significance of the results (see the caption of Fig. 8).

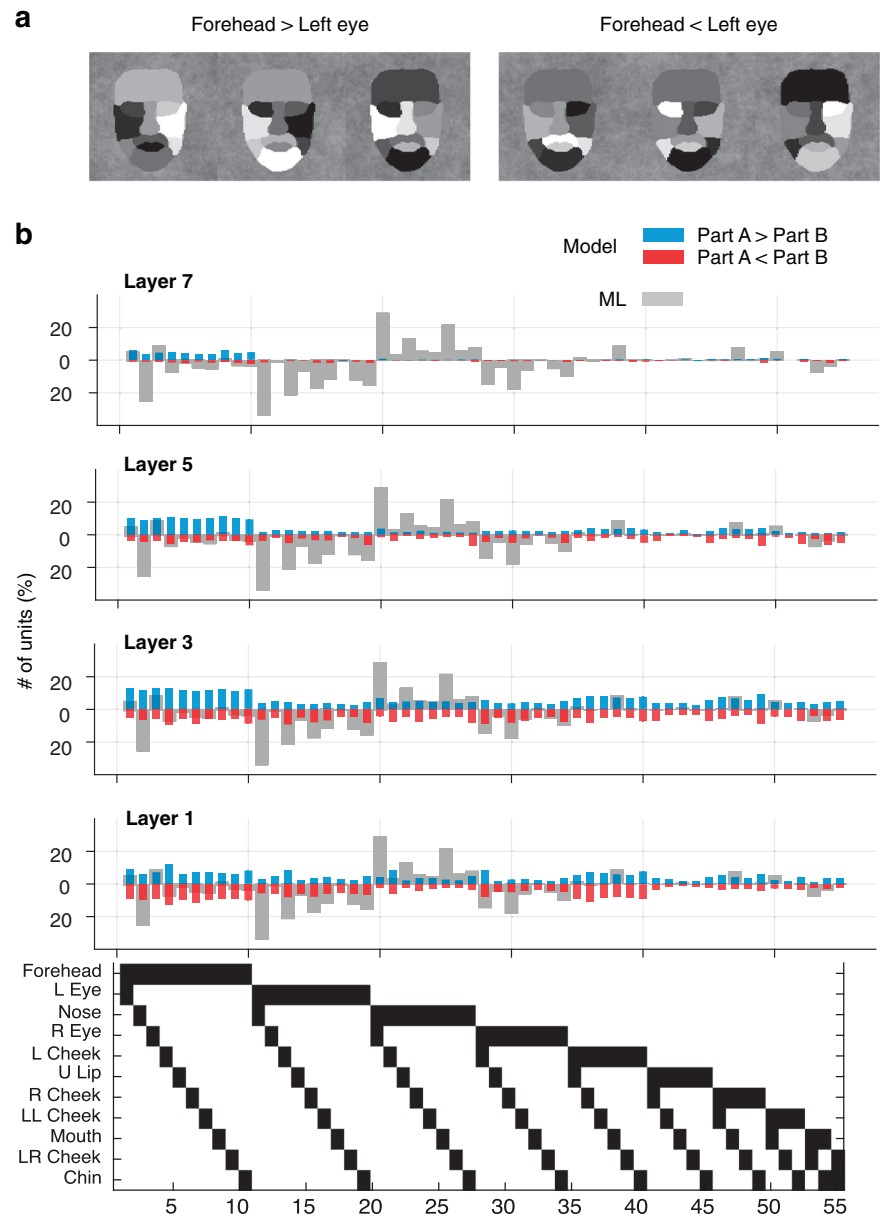

**Fig. 7 Contrast polarity tuning. a** Examples of mosaic-like cartoon face images with various intensity assignment to each face part. The first three have a larger intensity on the forehead than the left eye; the last three have the opposite. **b** The distribution of contrast polarity preferences in each model layer (blue and red) in comparison to ML (gray) replotted from Fig. 3A of the experimental study[17]. In each plot, the upper half gives the positive polarities (part A > part B), while the lower half gives the negative polarities (part A < part B). The binary table at the bottom indicates the 55 part-pairs; for each pair, the upper black block denotes A and the lower black block denotes B.

Comparing between layers, AM data favor higher model layers consistently across different tuning properties, namely, view and size invariance as well as appearance tuning (Fig. 8a–d). However, ML data favor different layers depending on each tuning property: intermediate layers in view-identity tuning (Fig. 8a), higher layers in size invariance (Fig. 8b), and lower layers in other tunings (Fig. 8c, e). Comparing between face patches, higher model layers also generally favor AM (Fig. 8a, c). However, intermediate layers are slightly inclined to ML in view-identity tuning (Fig. 8a) but to AM in shape-appearance preference (Fig. 8c). In sum, AM clearly corresponds with higher layers, while ML has no such clear correspondence since no layer is simultaneously compatible with all the compared experimental data on ML. (Note, however, that our study is confined to layer-

wise comparison; it remains open whether non-layer-wise correspondence exists for ML; see "Discussions".)

**Model variation.** How much robust are the results so far against the training condition? To address this question, we investigated various model instances while varying the architecture and the dataset.

First, we examined three publicly available pre-trained networks: (1) VGG-Face network[21], a very deep 16-layer CNN model trained on face images, (2) AlexNet[19], trained on general natural images, and (3) Oxford-102 network, an AlexNet-type model trained on flower images ("Methods"). Fig. 9 summarizes the layer-patch comparisons for all three models (shown in

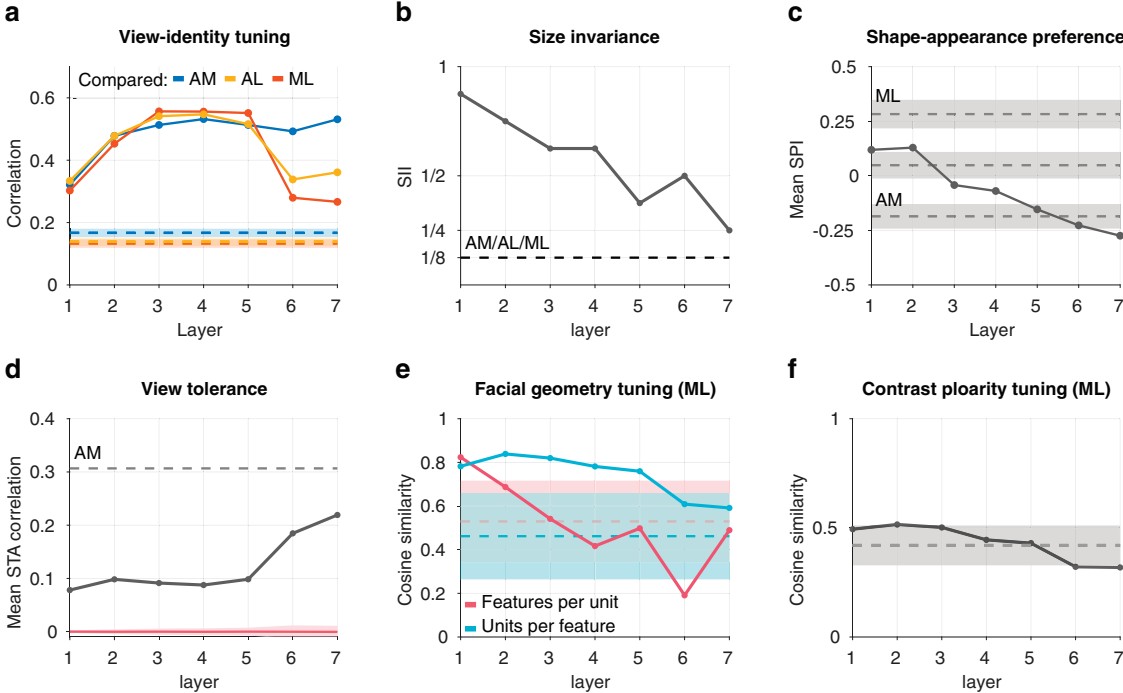

**Fig. 8 Summary of comparison between layers of AlexNet-Face and face-patches. a** The correlation between the RSM from each layer (Fig. 2) and each face patch (AM/AL/ML; Fig. 4d–f of the corresponding experimental study[15]). Each shaded region shows the ±2SD range of correlations from random cases, i.e., correlations between the experimental RSM and repeatedly generated random RSMs ("Methods"). **b** The size invariance index for each layer (Fig. 3b) and for face patches (equal for AM/AL/ML; Fig. S10C of the corresponding experimental study[15]). **c** The mean shape-preference index for each layer (Fig. 4b) compared with the mean indices for AM, ML, and their midpoint (estimated using Fig. 1e of the corresponding experimental study[18]). Each shaded region shows 95% confidence intervals constructed by 200 iterations of bootstrapping on the experimental data ("Methods"). Note that the mean SPIs for layers 1 to 4 exceed this interval for the midpoint. **d** The mean STA correlation for each layer (Fig. 5b) and AM (Fig. 6D of the corresponding experimental study[18]). The shaded region shows the ±2SD range of mean correlations between random STA vectors for the same population size as each layer ("Methods"). **e** The cosine similarity between the distributions of the number of tuned features per unit (red) or the number of tuned units per feature (blue) for each layer (Fig. 6) and ML (Fig. 3 of the corresponding experimental study[16]). **f** The cosine similarity between the distributions of contrast polarity preferences for each layer (Fig. 7b) and ML (Fig. 3A of the corresponding experimental study[17]). In **e**, **f**, each shaded region shows the ±2SD range of cosine similarities between the experimental distribution and randomly generated random distributions ("Methods").

different line styles), overlaid with the plots for AlexNet-Face (Fig. 8) and for an untrained AlexNet-type network. For VGG-Face network, we selected the layers that had the closest receptive field sizes as the layers of AlexNet ("Methods"). The results from these four trained models are overall similar: AM data tend to match with higher layers, while ML data do not match with any particular layer simultaneously for all the tuning properties. Surprisingly, some extent of consistency can be found in the models trained with non-face images (AlexNet and Oxford-102); however, the tendency is overall weaker, in particular, view tolerance (Fig. 9d), confirming the importance of training with face images. The different strength of size invariance in the top layer of different models (Fig. 9b) likely reflects the variety in image size in the training dataset (AlexNet-Face used data augmentation for size, while VGG-Face model did not; other models used a dataset that had size variation in itself). The results from the untrained model are, as expected, generally far from the face patches, though some are surprisingly similar (Fig. 9c, e), possibly due to the local computation inherent in the convolutional architecture (see "Discussions").

Second, to further explore the architecture space, we modified the AlexNet architecture to construct six additional networks, where four networks had five, six, eight, or nine layers and two networks changed the number of convolution filters in every layer to either half or double (Supplementary Table 1). We trained each model on the same face image dataset ("Methods"). Fig. 10 summarizes the results from these six models in addition to

AlexNet-Face, where the layer numbers are normalized. Again, the general tendency is similar across the architectures: AM corresponds to higher layers but ML has no corresponding layer. The weak view tolerance for the model with five layers or with half numbers of filters (Fig. 10d) is probably because the depth or the filter variety was not sufficient for gaining strong invariance.

## Discussions

In this study, we have investigated whether CNN can serve as a model of the macaque face-processing network. While simulating four previous physiological experiments[15–18] on a variety of CNNs trained for classification, we examined whether the results quantitatively match between each model layer and each face patch. As a result, higher model layers reasonably replicated the multiple tuning properties of AM, notably, strong invariance properties in size and view. Although such invariance properties in CNN are generally well-known as the model is trained to classify size- or view-varied images as the same, our finding in the fine-grained similarity to physiology in the view-tolerant appearance code goes beyond expectation (Fig. 5b). On the other hand, none of the CNN layers simultaneously captured those properties of ML: either a lower, intermediate, or higher layer showed the best match with ML for each different property. These observations were largely consistent across the model variation. Thus, despite the prevailing view linking CNNs and IT, a clear layer-wise correspondence seems to exist for face processing only in the last stage, not in the intermediate stage.

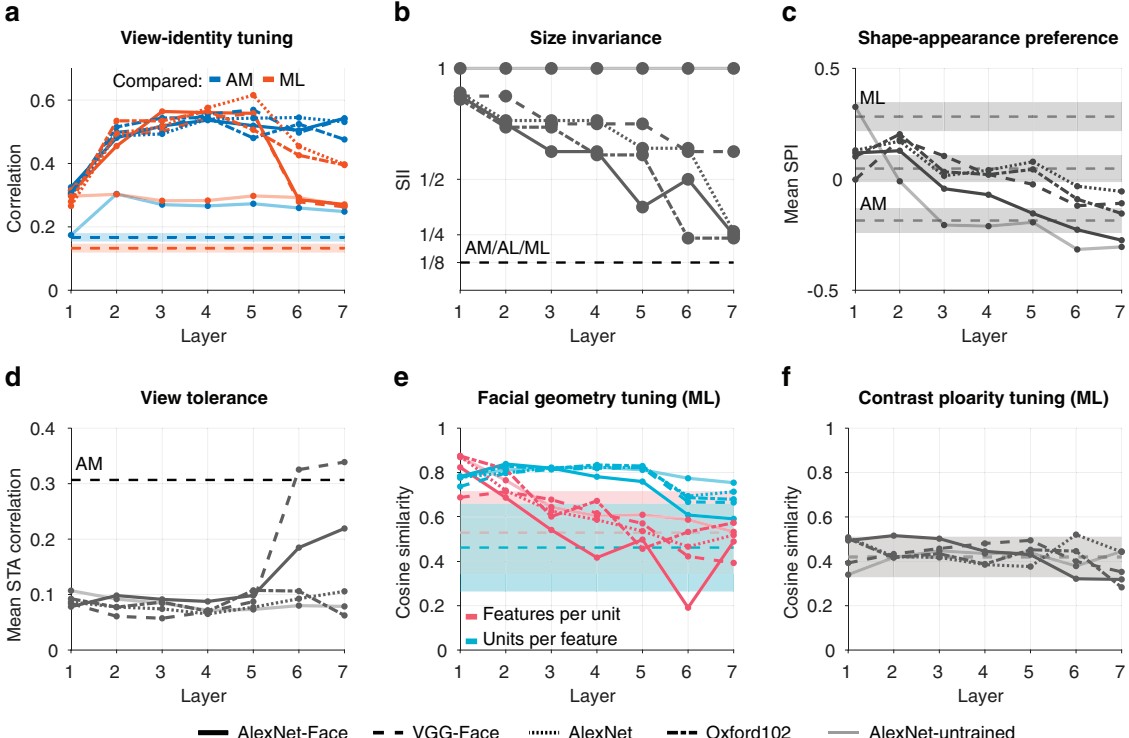

**Fig. 9 Summary of layer-patch comparisons for pre-trained and untrained networks.** The format of each plot (**a**–**f**) is analogous to Fig. 8 (omitting ±2SD regions in **d** due to the architecture variety). The networks including AlexNet-Face are indicated in different line styles (see the legend at the bottom). In the view-identity tuning plot **a**, we omit comparison with AL data for visibility. In the size invariance plot **b**, we slightly shift each curve vertically also for visibility. For VGG-Face, the seven layers were those with the closest receptive field sizes to the corresponding layers in AlexNet.

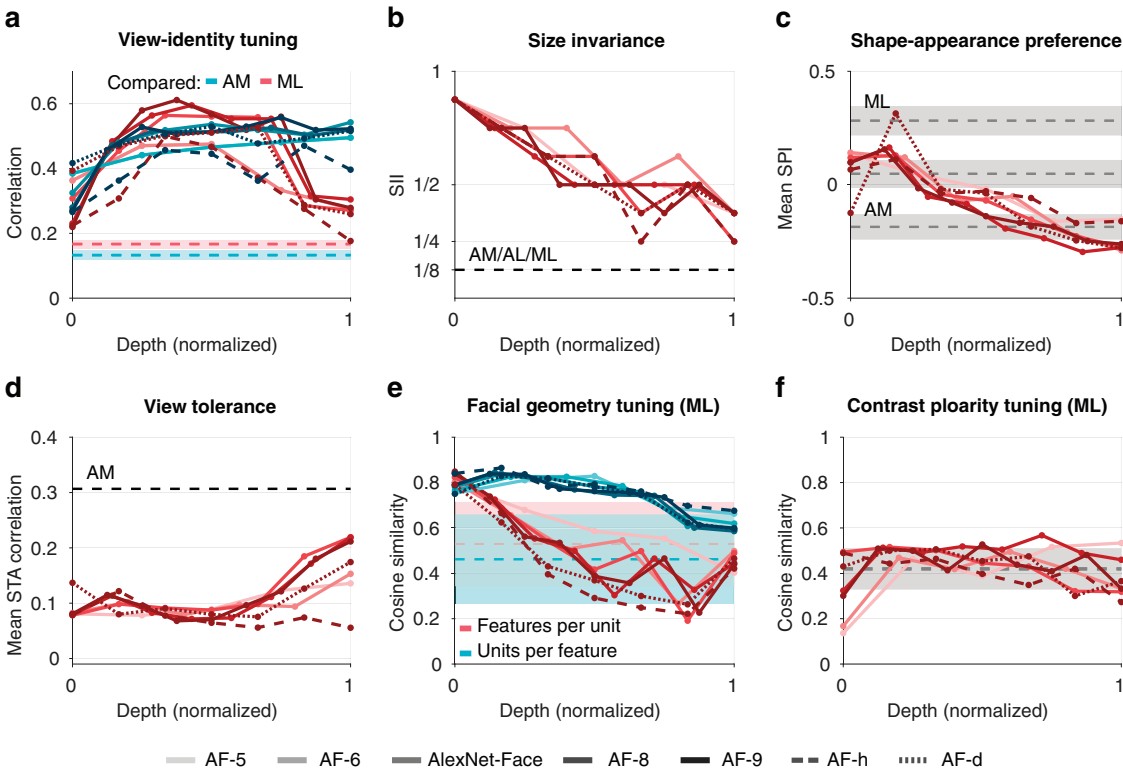

**Fig. 10 Summary of layer-patch comparisons for different architectures.** The format of each plot (**a**–**f**) is similar to Fig. 8, except that x-axis shows the normalized depth (0 corresponds to the lowest layer and 1 to the highest layer). The architectures differ in the depth or the number of convolution filters, indicated by different colors or line styles; AF-5 to 9 varied the depth; AF-h halved and AF-d doubled the number of filters in each layer (see the legend at the bottom).

However, our approach crucially relies on the assumption that each previous monkey study investigated a cell population that was sampled from the entire target face patch without much bias. However, since there is no supporting or contradicting evidence on this, we cannot completely reject the possibility that the multiple properties of ML could actually have been of different sub-clusters of ML. Thus, despite our failure in finding a precise layer-to-patch correspondence for ML, it still remains possible that some other more complicated correspondence might exist, e.g., combined multiple layers corresponding to a single face patch.

Some previous studies have also compared CNN and face patches for relating it with their main experimental findings or computational model. One study[18] found AM-like shape-appearance tuning in the top layer of a face-classifying CNN model (appearance preference and ramp-flat tuning), similarly to our results (Fig. 4b and Supplementary Fig. 5, layer 7). Another study[22] tested view-identity tuning on several CNN models to compare with their novel generative model (see below). Although they showed RSM results similar to ours (Fig. 2), they incorporated a more sophisticated quantitative comparison with experimental data[15] and thereby revealed notable similarity in the later stage and dissimilarity in the intermediate stage, which is generally compatible with our conclusion. Thus, these studies have somewhat anticipated our perspective on CNNs in relation to the face patches. However, our study has added a substantial assurance by taking a much larger variety of face-patch experiments and CNN models into consideration.

What insight did we gain from our results? First, appearance representation is dominant in later stages in both systems (Fig. 4), whereas shape representation lacks in intermediate-to-higher stages in CNN (Figs. 4 and 6) and is dominant only in intermediate stages in macaque. This may be because shapes are in fact relatively unimportant features for classification and thus neglected during the model training[23]. This implies that the goal of the face-processing system may not be merely classification. Second, lower-layer units often showed somewhat unintuitive properties despite that they were expectedly less face-related: (1) significant facial shape tuning (Figs. 4b and 6); (2) higher decoding performance of facial features than higher layers (Supplementary Fig. 4), and (3) ramp-shape tuning along STA axis and flat tuning along the orthogonal axis, not limited to higher layers (Supplementary Figs. 5 and 7). These may be partly because lower-layer units, although simple, localized feature detectors (e.g., Gabor or random filters), can in fact easily interact with stimulus parameters controlling shape feature dimensions or local facial geometry. Also, lower layers have much weaker nonlinearity so that linear decoding would become easier. One could argue that such low-level units should not be qualified as face-selective from the first place, but might have been misjudged so by the standard criterion due to the specific image statistics of face images (e.g., emphasis on eyes). This point also suggests that each single experiment generally has limitation in discriminating between plausible and implausible models, which underscores the importance of taking multiple experiments into account.

In general, relationship between artificial and biological neural networks has been a recurring question. Since the brain has a number of sub-networks with a hierarchical structure, it is tempting to hypothesize that such sub-network is optimized for some behavioral goal. Indeed, a classical study has shown that a neural network trained for a coordinate transformation task exhibits, in the intermediate layer, properties related to spatial location similar to primate parietal area 7a[24]. As mentioned in Introduction, more recent studies have argued that CNNs trained for image classification have layers similar to higher[2–4,7], intermediate[3–5], or lower[6] areas in the monkey or human visual

ventral stream. Analogously, layer-wise correspondence has been found between CNNs trained for audio classification and the human auditory cortex[25] or the monkey peripheral auditory network[26]. Although all these studies are positive in the generality of explanatory capabilities of goal-optimized neural networks, the same story might not go all the way through. For the macaque face-processing network, our study here showed that correspondence with CNN should at least not be layer-wise. In addition, some recent studies have pointed out potential representational discrepancies between CNN and the ventral stream from behavioral consideration[22,23,27–29]. (See also the related discussion on the "computational gap" below.)

One should note that there are, in general, two fundamentally different approaches for comparing a model with the neural visual system. In one approach, which is more traditional and taken in our study, a given model is tested whether it exhibits tuning properties compatible with prior experimental observations ("tuning approach"). In the other approach, which is recently more popular[2,3,25], a given model is used as a basis function and a linear regression fitting is conducted from model responses to actual neural responses for predicting new neural responses ("fitting approach"). In the tuning approach, although the comparison is arguably more direct in the sense of involving no fitting, tuning experiments have often been criticized for biased and subjective stimulus design and for use of degenerate summary statistics. Therefore showing consistency with experimentally observed tuning may not be sufficiently supportive evidence for the model. Note that, nevertheless, showing clear inconsistency is strongly falsifying evidence, from the logical contraposition of "if the model behaves similarly to the neural system, then it should reproduce a similar tuning property." Our comparison with ML would be one such example. In the fitting approach, on the other hand, the aforementioned criticism would not occur since an arbitrary (randomly selected) set of stimuli can be used. However, the necessity of linear fitting makes the comparison somewhat more indirect: correspondence is made between an actual neuron and a "synthetic neuron," i.e., the output of a linear model after fitting[2–4,30]. Indeed, one example of computational gap between CNN and IT neural responses has been raised in Fig. 7 of the study by Cadieu et al.[2], where the population similarity matrices from IT and a CNN top layer were strikingly different without fitting, although very similar with fitting. One might therefore argue that it is a CNN plus a linear regression, not a CNN itself, that has predicted neural responses. Thus, both tuning and fitting approaches are complementary to each other and neither is significantly better than the other.

If CNN does not fully explain all the facial tuning properties, then what can be alternative models? Some hints can be found in prior theoretical studies. First, although unsupervised learning of feature representations from the image statistics has traditionally been used for early vision[31–34], recent studies have raised its possible contributions in facial tuning properties. For example, sparse coding of facial images can produce facial-part-like feature representations, which explains surprisingly well most of the facial geometry tuning properties found in ML[35]. Also, PCA-based learning of face images can produce global facial feature representations, which exhibit monotonic and mirror-symmetric view tuning as in ML and AL[36]. Second, feedback processing is ubiquitous in the visual system and therefore likely important, but crucially missing in CNN. One standard theoretical approach to incorporate feedbacks is to use a generative model. Although such theory has also been typical for modeling early vision[37–39], it can potentially be important in higher vision. For example, a particular generative model, called a mixture of sparse coding models, has used multiple modules of feature representations with competitive interaction, which can endow model units with

a global face detection capability similar to the neural face selectivity[35]. In another approach, a novel generative model assumes a computer graphics algorithm that generates face images from certain facial and scene feature parameters, while employing a feedforward deep convolutional network trained on those feature parameters[22]. Notably, certain three layers in the feedforward network exhibited view-identity tuning properties each similar to ML, AL, and AM in a way quantitatively better than standard CNN models[22]. In a different approach, deep networks added with recurrent connections have recently been used to explain late-phase neural dynamics in IT[40,41]. Third, invariance properties can be explained not only by supervised learning as in CNN but also by image statistics. While spatial statistics can explain well position or phase invariance in early vision[32,42,43], temporal coherence[44] for learning the most slowly changing features has been commonly used for explaining more complex invariance properties in higher vision[36,45,46] and experimentally tested[47]. Although combining such invariance learning into a generative model is theoretically not so obvious, one approach extending variational autoencoders[48] has been developed[49] and led to a novel deep generative model explaining multiple tuning properties in ML and AM[50]. Taken together, for clarifying the computational principle underlying the primate face-processing system, it seems crucial to view this system as not merely a classifier but having a richer repertoire of visual processing, for which the present work would offer a solid motivation.

## Methods

**Convolutional neural network**. CNN is a family of feedforward, multi-layered computational models[51], which has originally been inspired by the mammalian visual system. CNN allows for a variety of architecture design, stacking an arbitrary number of layers with different structural parameters. Each layer in CNN undergoes several operations. The layer typically starts with convolutional filtering, which applies an identical multi-channel linear filter to every local subregion throughout the visual field. Then, the results are given to a nonlinear function, $ReLU(x) = \max(0, x)$, at each dimension, for ensuring non-negative outputs. The layer is optionally preceded by pooling and normalization. Pooling takes the maximum of the incoming inputs within a local spatial region. Normalization divides the incoming inputs by their squared norm. Generally, a CNN architecture consists of multiple such layers, by which it progressively increases the effective receptive field sizes and eventually achieves a non-linear transform of the input from the whole image space to a space of interest (i.e., class). Often, the last several layers of a CNN architecture have convolutional filters covering the entire visual field, thus called fully connected layers.

Each layer operation is closely related to some neural computation discovered in neurophysiology. Convolutional filtering mimics V1 simple cells, which replicate their receptive field structures across the visual field[52]. The nonlinear function proxies for neural thresholding giving rise to non-negative values of firing rates. Pooling comes from the classical notion that V1 complex cells gather the outputs from V1 simple cells to achieve position or phase invariance[52,53]. Normalization stems from a gain-controlling phenomenon that is widely observed in the cortex and often explained by the well-known divisive normalization theory[54]. Further, the repetition of layers of similar processes is inspired by the hierarchy of the visual cortex, for which the gradual increase of receptive field sizes and the congruent micro-circuit structures are well known[55]. To dive deep in CNN, see introductory materials[56,57].

**Trained CNN models**. We show, most in detail, the results from a representative CNN model called "AlexNet-Face." This CNN model has the same architecture as AlexNet[19] with five convolutional layers followed by two fully connected layers. (The network ends with a special layer for representing classes, but we ignore it in our analysis.) The architecture parameters are given in Supplementary Table 1. We trained it for the classification task using the VGG-Face dataset[21], which contains millions of face images of 2622 identities. We augmented the dataset with size variation, allowing four-times downsizing. (Note that four-times downsizing was limit in our case with full image size 224 × 224 since further downsizing would make the images too small and impossible to discriminate and thus considerably degrade the classification performance of the model.) We performed the training by minimizing the cross-entropy loss function, a commonly used probabilistic approach to measure the error between the computed and given outputs[56]; we used the stochastic gradient descent method with momentum (SGDM) as optimizer. The resulting CNN model gave classification accuracy 72.78% for held-out test data. (This score is somewhat lower than state-of-art face recognizing deep nets, which typically go over 90% of accuracy. This is likely because our size-varied data augmentation yielded very small images that would be difficult to classify, e.g., Fig. 3a).

To test robustness of our results against structural change to the model, we incorporated a set of six additional model instances modifying the architecture of AlexNet. Four of them changed the number of layers to five, six, eight, and nine. We designed the specific architectures for these by changing only the convolutional layers, keeping the overall structure of increasing receptive field sizes. The remaining two models changed the number of filters in every layer, one halved and one doubled. The architecture parameters of the additional models as well as their classification accuracies are given in Supplementary Table 1.

For all implementation, we used Matlab with Deep Learning Toolbox (https://www.mathworks.com/products/deep-learning.html) as well as Gramm plotting tool[58] for visualization.

**Pre-trained CNN models**. To further test robustness, we included three publicly available pre-trained CNN models using very different architecture or dataset, namely, VGG-Face network[21], AlexNet[19], and Oxford-102 network. The VGG-Face network is a very deep 16-layer CNN model that has been trained on VGG-Face database for face classification (with no data augmentation for size variation). For analysis of the VGG-Face network, we chose the layers that had the closest receptive field sizes as the layers of AlexNet (Supplementary Table 2). Also, since lower layers of VGG-Face were too large, we analyzed a subpopulation of randomly sampled 30,000 (face-selective) units whenever the full population exceeded this number. AlexNet is the well-known, original network trained on ImageNet database[59] for natural image classification. Oxford102 is an AlexNet network that has been 'fine-tuned' for the classification of flower images in Oxford-102 dataset[60]. We imported these three network models from a public repository (https://github.com/BVLC/caffe/wiki/Model-Zoo).

**Experiments**. On each CNN model, we first identified face-selective units and then proceeded to simulation of four macaque experiments on these face-selective units. The specific procedures are summarized as follows.

**Face-selective population estimation**. We determined the face-selectivity of a unit by following the general approach used in experiments on IT[15,16]. That is, we first recorded the responses of the unit to a set of 50 natural frontal faces from the FEI image database (http://fei.edu.br/~cet/facedatabase.html) and to a set of 50 non-face object images obtained from the Web. Then, from the average responses to the faces, $\bar{R}_{face}$, and non-face object images, $\bar{R}_{object}$, above the baseline response to the blank image (all zero pixel-values), we estimated the Face Selective Index, $FSI = (\bar{R}_{face} - \bar{R}_{object})/(\bar{R}_{face} + \bar{R}_{object})$. FSI was set to 1 when $\bar{R}_{face} > 0$ and $\bar{R}_{object} < 0$, and to $-1$ when $\bar{R}_{face} < 0$ and $\bar{R}_{object} > 0$. We judged a unit as face-selective if $FSI > \frac{1}{3}$, that is, the unit responded to face images, on average, twice as strongly as to non-face object images. (Zero FSI, for instance, implies an equal average response to face and non-face images.) For the simulation of shape-appearance tuning experiment described below, we used profile faces in addition to frontal faces for selectivity determination. Below, 'unit' or 'population' always refer to the face-selective ones.

**View-identity tuning experiment**. To simulate the experiment on view-identity tuning[15], we used the same set of "face-view" (FV) images as in the experimental study, which consisted of 200 images of 25 identities and 8 views (left full-profile, left half-profile, frontal, right half-profile, right full-profile, up, down, and back; see the images along the axes in Fig. 2). To determine the view-identity tuning, we first recorded the responses of the units to these images and calculated the correlation between the population responses to each pair of FV images; we then constructed a population response similarity matrix (RSM) from those values. The authors of the experiment provided us the FV image set and the RSMs obtained from the experiment (the data corresponding to the back view was missing). We quantified the similarity between a model RSM and an experimental RSM by their correlation coefficient and tested whether it exceeded the ±2SD range of the correlation coefficients between the experimental RSM and repeatedly generated random RSMs, i.e., symmetric matrices whose diagonals are all one and non-diagonals are drawn from Gaussian distribution with the mean and variance matched to the experimental RSM.

**Size-invariance experiment**. To simulate the experiment on size invariance[15], we took frontal as well as right and left profile face images of 8 individuals (24 face images in total) from FEI face database, and 16 non-face object images obtained from the Web. We then resized these images from the original size 224 × 224 pixels down to 196 × 196, 168 × 168, 140 × 140, 112 × 112, 84 × 84, and 56 × 56 pixels (see Fig. 3a), forming a set of 280 images in total. For each layer, we recorded the responses of the units to those images and calculated the average across the population and the image set, $\bar{R}_{face}^{size}$ or $\bar{R}_{object}^{size}$, separately for faces or objects, and separately for each size. To quantify the degree of size-invariance, we define size-invariance index (SII) as the minimal fraction of sizes at which the mean

population response to faces is reasonably larger than objects: ($\bar{R}_{face}^{size} > 1.4\bar{R}_{object}^{size}$). (If no image size gives the required level of face preference, SII is defined as 1.) We compared the SIIs of the model layers and the face patches, where all ML, AL, and AM give around the value 1/8, according to the experimental data (Fig. S10 of the experimental study[15]).

**Shape-appearance tuning experiment**. To simulate the experiment on shape-appearance tuning[18], we followed their approach to create a face space based on the active appearance model[20]. For this, we employed 200 frontal face images from FEI face dataset, on which we annotated the coordinates of pre-defined 95 facial landmarks. We here used the Face++ tool (https://www.faceplusplus.com) to automatically annotate the landmarks supported by this tool and resorted to manual annotation for the other landmarks. Our set of landmarks were in fact a superset of the ones used experimentally (58 landmarks) since the automatic tool happened to provide richer annotations, which would not (at least) impoverish our result.

From the above annotated image dataset, we constructed a 50-dimensional face space as follows. We first performed principle component analysis (PCA) on the landmark coordinates of the 200 images, of which the first 25 principal components provided the first 25 dimensions of the face space, called shape dimensions or features. Thereafter, we normalized each image by morphing the original image so that the warped landmark coordinates match the mean landmark coordinates (across all images). We again performed PCA on the resulting normalized images, of which the first 25 principal components provided the last 25 dimensions of the face space, called appearance dimensions or features. We then generated a set of 2000 face images by randomly sampling feature vectors from the 50-dimensional isotropic Gaussian distribution and reconstructing images by reversely following the above process. Fig. 4a shows examples of generated frontal face images by varying the first shape and appearance dimensions.

To identify shape-appearance preference of each unit, we first recorded the responses to the generated 2000 face images and calculated the STA as the average of the 50-dimensional feature vectors of those images weighted by the corresponding responses. To quantify the preference to shape or appearance, we defined shape preference index (SPI):

$$\text{SPI} = \frac{(S - A)}{(S + A)}$$

where $S$ is the vector length of the 25 shape dimensions and $A$ is the vector length of the appearance dimensions of the STA[18]. To compare the population distributions of SPIs for the model layers and the face patches, we used their mean SPIs. We also tested whether the mean SPI from each model layer exceeded the 95% confidence intervals constructed from 200 bootstrap samples of the experimental data, which were extracted from Fig. 1E of the experimental study[18].

To investigate view tolerance, we repeated the aforementioned face space generation process for the profile view. That is, we created a profile face space from 200 profile face images from FEI database with manually annotated landmark coordinates and randomly generated 2000 profile face images. We established a mapping between the frontal and profile face spaces by linear regression between the frontal and profile feature vectors of the same identities. From then on, we always used the profile feature vectors mapped to the frontal face space. Fig. 5a shows examples of generated left profile face images by varying the first shape and appearance dimensions. We thereafter calculated the STA for profile face images similarly to frontal face images. We quantified view tolerance by correlation between the corresponding STA dimensions for frontal and profile images across units. We also repeated the same process for all remaining non-frontal views (10 views in total). See the original experimental study[18] for details. We compared the model and experimental results by the mean of the STA correlations and tested whether it exceeded the ±2SD range of mean correlations between random STA vectors (drawn from standard Gaussian distribution) for the same population size as each layer.

**Facial geometry tuning experiment**. To simulate the experiment using cartoon face stimuli[16], we used the same cartoon face design, where each face image had seven elementary parts (hair, face outline, eyes, irises, eyebrows, mouth, and nose) and the geometry of the parts were parameterized by 19 feature parameters (face aspect ratio, face direction, height of assembly, hair length, hair thickness, eyebrow slant, eyebrow width, eyebrow height, inter-eye distance, eye eccentricity, eye size, iris size, gaze direction, nose base, nose altitude, mouth-nose distance, mouth size, mouth top, and mouth bottom). Each of these parameters held values between −5 to +5 (11 values), where zero corresponded to the average features (e.g., normal inter-eye distance) and ±5 corresponded to the extreme features (e.g., large or no inter-eye distance). Fig. 6a shows example cartoon face images that vary the inter-eye distance parameter over 11 different values.

Using the same method as in the experimental study[16], we estimated a tuning curve for each unit and for each feature. For this, we first generated a set of 5000 cartoon face images with random values for the 19 parameters. From the responses to those images, we estimated a tuning curve for each feature parameter by taking the average of the responses corresponding to each value that the feature parameter takes, regardless of the values of the remaining features; we smoothed the curve by a Gaussian kernel of unit variance. We then determined the statistical significance of the tuning curve using the same criterion as in the experimental study[16]. We examined the population distributions of the number of significantly tuned FPU and the number of tuned UPF. For both, we quantified the similarity between the model and experimental distributions by taking their cosine similarity. We tested whether the similarity exceeded the ±2SD range of cosine similarities between the experimental distribution and repeatedly generated random distributions. Here, a random distribution for FPU was generated by assuming each unit having a random number of tuned features drawn from the uniform distribution ranging from 0 to 19; a random distribution for UPF was generated by assuming each feature having a random number of tuned units drawn from a uniform distribution ranging from 0 to 100.

**Contrast polarity tuning experiment**. To simulate the experiment using mosaic-like parameterized cartoon faces images[17], we used the same stimulus design, where each face image consisted of 11 face parts (forehead, left eye, right eye, nose, upper lip, left cheek, right-cheek, lower-left cheek, lower-right cheek, mouth, chin) based on a manual fragmentation of a mean face. Following the experimental procedure[17], we generated a set of 432 faces images by randomly giving a unique intensity value to each part ranging between dark and bright (11 values), while ensuring that, for every possible part-pair (55 part-pairs in total, e.g. forehead and left-eye, forehead and right-eye) and every possible intensity level, the set included at least one exemplar in which that part-pair had that intensity level. Fig. 7a shows some examples of generated mosaic-like cartoon face images in the case of forehead lighter than left eye or the opposite.

Following the experimental method[17], we used the responses of each unit to the above set of images to determine preference on the contrast polarity for each of the 55 part-pairs. More specifically, for each part-pair (A and B), we compared the average response between the condition where intensity of part A was greater than part B (part A > part B) and the condition where intensity of part A was lesser than part B (part A < part B) irrespective of what intensity value the remaining 9 parts held. We then determined the significance of the contrast polarity preference for each part-pair using the same criterion as in the experimental study[17]. We examined the population results in terms of the distributions of the number of significant preferences per part-pair. We quantified similarity in the same way as facial geometry tuning (UPF) described above.

**Statistics and reproducibility**. The statistical methods used in this study are described above. All the results shown here are reproducible with the specified datasets and parameters.

**Reporting summary**. Further information on research design is available in the Nature Research Reporting Summary linked to this article.

## Data availability

The authors declare that all data supporting the findings of this study are available within the paper and its supplementary information files. All relevant data will be available upon request. Requests can be made to the corresponding author.

## Code availability

The code for training network models, simulating past experiments, and generating figures was written in Matlab (2018b) with Deep Learning Toolbox and Gramm plotting tool. The code is available publicly: https://github.com/HaruoHosoya/cnn_face.

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

## Acknowledgements

This work has been supported by the Commissioned Research of National Institute of Information and Communications Technology (1940201), the New Energy and Industrial Technology Development Organization (P15009), and Grants-in-Aid for Scientific Research (18H05021, 18K11517, and 19H04999).

## Author contributions

H.H. designed the study. R.R. and H.H. created software required for the study. R.R. performed simulation and analysis. R.R. and H.H. wrote the manuscript.

## Competing interests

The authors declare no competing interests.
