## [Peer Review File · Communications Biology]

Reviewers' comments:

Reviewer #1 (Remarks to the Author):

In an effort to better understand the neural computations underlying face perception, the paper presents a detailed comparison of popular convolutional neural networks with several well-known neurophysiological findings about the macaque face patch system. Specifically, it compares the tuning properties of the model units, considering each layer in a given network, to the tuning properties of neurons across three levels in the hierarchy of the face patch system (MLMF, AL, AM), which have been previously reported across four electrophysiology studies in the literature. The authors characterize their results as follows: higher CNN layers appear to correspond well to the most high-level, anterior face processing areas, but earlier CNN layers did not systematically correspond to the middle face processing area. They conclude that even though the near-goal final stages of CNN layers has some similarities with the primate face system, the intermediate stages of processing in the system may be very different than that of CNN's.

The extent of comparisons presented in the paper as well as the rigour with which they are carried out are impressive and make the paper worthy of publication. These results should establish a point of reference for future studies looking to develop more faithful computational theories of primate face processing system. However, the paper shouldn't be published in its current form: there are some problems with the authors' core argument and conclusion that are fixable but must be fixed before it can be published. We could support its publication in *Communications Biology* if the authors can carefully address the issues we describe below.

For full disclosure, we are co-authors of a paper that represents importantly related work, which is cited in a number of places in the current paper. It is impossible to review the paper without acknowledging this is our perspective, and we believe this doesn't stand in the way of us giving a fair, impartial, and indeed appreciative review of the present work.

1. What conclusion do the analyses really support?

Their presented analyses do not, at least in the simple way the paper suggests, support one of the paper's main conclusions: that higher CNN layers can explain anterior regions while intermediate CNN layers cannot explain middle regions. Part of their evidence for this conclusion is that the latest stage (layer 7) in Alexnet-face seems to give the best fit (or the second best fit) of the anterior face processing region AM across all datasets that they looked at. They then observe that this is not the case for ML/MF: sometimes it is best fit by layer 1 and sometimes it is better fit by higher layers (4 or 5); because there is more variability in terms of the Alexnet layer that best fits ML/MF from dataset to dataset, the authors conclude that it ML/MF is not as well explained by intermediate layers of convnets.

However, it seems that we could conclude something different -- indeed, we could conclude the opposite of what the authors conclude -- if we focused on the VGG-Face network results rather than AlexNet. Looking at Fig. 9: (1) Across the board, in the VGG-face network, there is considerable consistency in terms of the layers (layer 4 and 5) that fit ML/MF best, which is comparable to what they report in Alexnet-face layers 6 and 7 vs. AM comparisons; and (2) As Fig. 9a shows, the VGG-Face network layer that fits both ML/MF and AM best (layer 5) correlates with ML/MF to higher degree relative to AM. This finding is in fact consistent with a report comparing VGG-face activations and ECoG recordings in humans (Grossman et al., 2018). Authors of this human neurophysiological study indeed conclude the opposite of the present paper: that intermediate stages of CNNs correspond well to how the brain processes faces, while later stages of CNNs do something different from what the

brain does.

So, at a minimum, the story is more complicated than the paper suggests in its conclusion. Further complicating the interpretation is this: Despite the fact that there are higher correlations between layer 5 in the VGG-Face network and ML/MF, as the authors richly demonstrate, all of those intermediate and final stages across all networks lack the kind of shape tuning that is reported in the literature (Chang & Tsao; Ohayon et al.).

Overall, we believe that a more fair conclusion would be to say that CNNs trained to recognize faces do show some similarities to the brain's mechanisms but also a number of pronounced differences. They give rise to view-invariant coding at their higher layers as in the anterior regions in the brain, but both the nature of that code and the way it is computed from images might be quite different from the brain. None of the existing, standard CNN models gives a good account of the entire face patch system in the brain.

2. Relation to literature

When the authors compare their work to our earlier work, their characterization seems to us not entirely fair. They write: "Some previous studies have addressed a similar question but in a smaller scale. In one study (Yildirim et al., 2018), the authors tested view-identity tuning on a CNN model in comparison to experimental data (Freiwald and Tsao, 2010) and gave a similar result to ours (Figure 2). In another study (Chang and Tsao, 2017), the authors trained a smallish face-classifying CNN model and tested shape appearance tuning."

They then describe some differences between their work and these earlier studies, and conclude: "Thus, the comparisons undertaken in our study here are far more thorough and extensive than those previous studies."

We believe it's not fair to say simply that the present study is "far more thorough and extensive" than either our previous work or Chang and Tsao's. Those previous studies looked at fewer datasets, but analyzed them more deeply -- and arguably, much more "thoroughly". Both our paper and Tsao's also presented additional, new data which the present paper does not consider. So the current work is importantly complementary to previous comparisons of CNNs and the macaque face patches, not strictly better. Its strength is to look more broadly at a wider range of previously published studies, which is valuable in a different and complementary way than the value of the earlier papers cited.

Another issue related to our work is how the authors characterize the new model we present, the "Efficient Inverse Graphics" or EIG network, and how it fits with their own interpretation of their findings. They argue -- at least, this is the part of their story we think is strongest (see earlier points of our review) -- that conventional CNNs capture some aspects of how the brain represents identity (or appearance?) in a viewpoint-invariant way, but not all aspects, and not how it gets there -- not the computations that lead to those viewpoint-invariant identity/appearance representations; also, conventional CNNs do not capture how the middle patches represent shape. That in a sense is also one of the core arguments in our paper; moreover, we present an alternative model -- an alternative kind of deep convolutional network, with a different training target, that is constructed to invert a graphics-like generative model, and which does capture these properties of the face patches (at least as represented in the Friewald and Tsao 2010 data) much better than standard CNNs. We have not yet tested our model on all the datasets that the present paper considers, which would be an important target for future work. The present paper inspires us to do that. But we believe it is important for the authors to acknowledge in a revised version of this paper that (1) the core argument

it can sustain is also consistent with the findings and the argument we have made previously, looking at only a subset of the data considered here; and also (2) that we have proposed at least one model which takes the form of a deep convolutional network and which could potentially explain the full face-patch circuitry better than standard CNNs from the computer vision literature, but which needs to be further tested.

3. More minor comments.

It sounds strange and unscientific to use the word “smallish” to characterize the dataset in the Chang and Tsao paper. There must be a better word to use here?

The authors don't comment much on AL, and its mirror symmetry properties which have been the subject of a number of other modeling efforts (e.g., by Leibo and colleagues). For example, in Fig. 2, they see some mirror-symmetry in an earlier region which goes away or weakens in higher stages, which runs against at least a simple conclusion that only higher stages match anterior regions.

The size-invariance analysis is interesting and deserves more emphasis and discussion. Why is there such a big discrepancy across the board -- ML/MF, AL and AM are all more size-invariant than the CNNs, even more so than their highest stages? What would it take for a computational model to match this constant and high-level of size invariance reported in the data?

In supplementary Fig S3, can the authors comment on why decoding accuracy decreases from layer 5 to layer 7? We found that surprising.

We were confused by the use of the word “explains”: It is not clear what it means to say that CNNs explain some part of the face patch system according to the paper. If the authors want to use that word, then they should clarify the nature of that explanation. Is it just about matching or fitting the data, or is there some deeper sense in which the authors think their analyses explain how the neural circuits are working? Alternatively, the paper could refer to “match” or “fit” between CNNs and data, which is the more concrete result that they demonstrate.

Thank you for giving us the opportunity to review this interesting paper, which we hope could make a valuable contribution to the literature.

Ilker Yildirim and Josh Tenenbaum

Reviewer #2 (Remarks to the Author):

Recently, the similarity between CNN and ventral visual pathway has been one of major issues in the cognitive neuroscience. For each layer, for each property, this manuscript provides extensive and detail comparison between the biological and artificial neural networks, useful for various researchers, especially for computational neuroscientist. In my opinion, this manuscript is worth to be considered to be published in Communications Biology. However, I hope to these comments are considered to be improve the readability of the manuscript.

Major comments:

1. At first, it is not clearly described how the results from “the past experiment” in Fig. 1 are acquired.

Did you (the authors) receive the results from the four different research groups? Did you receive the original neural responses? or just copy the values from the published papers? How can you sure that the neural responses from the past experiments and node activations from CNN are treated in the same way to draw the figures? Can we directly compare the neural responses and node activations without any normalization or fitting process? Explanations or references are required.

2. The authors used “mean STA correlation” from shape-appearance model (Chang and Tsao, 2017) to compare the view tolerance. This measure is the specific result from the shape-appearance model, but not direct measure to evaluate the view tolerance. In my opinion, the direct way to quantify the view tolerance would be to examine identity and view tuning curves, and compare the ratio between the variance across identities and the variance across views. I recommend to focus on discovering the actual difference between the ventral visual pathway and CNN, rather than reproducing the results from the previous works.

3. Overall, this manuscript is well written, and the results look clear. However, the contents related to the size invariance and Fig. 3 (especially, Fig. 3B) are not easy to understand. To my understanding, the size invariance should be revealed as the “flat” tuning to the stimuli with different sizes. However, in Fig. 3B, the shapes of tuning curves look similar regardless of different layers. The only difference that I can find is the smaller average responses in higher layers. Why the smaller responses mean the higher invariance? In addition, the sentence “We then calculated, for each layer, the average responses, ... for each image size” is hard to read.

Revision of manuscript "CNN explains tuning properties of anterior, but not middle, face-processing areas in macaque IT"

We would like to thank the reviewers for very useful comments. We have made a major revision of our previously submitted manuscript.

Reviewer #1 was mainly concerned with the appropriateness of the conclusion and the relationship with existing model studies. To address this, we have undertaken redoing of a whole analysis of the CNN model that the reviewer questioned on as well as major rewriting in Introduction, Results, and Discussion sections. In addition, the reviewer raised a number of points that were not clear enough in our initial manuscript. We made the best efforts to clarify every point.

Reviewer #2 also pointed out several points that needed clarification. We also dealt with each point with a great care.

Incorporating the comments by both reviewers led to a major improvement of the manuscript. The modifications in the revised manuscript are highlighted by blue font. We also made a lot of edits without changing the meaning to meet the required paper length (not highlighted).

Below are our point-by-point replies to the comments. Each individual comment was given numbers, e.g., (1-2) indicates the first reviewer's 2nd comment, in red font and our reaction to each comment was described in black font.

Reviewer #1:

(1-1)

1. What conclusion do the analyses really support?

Their presented analyses do not, at least in the simple way the paper suggests, support one of the paper's main conclusions: that higher CNN layers can explain anterior regions while intermediate CNN layers cannot explain middle regions. Part of their evidence for this conclusion is that the latest stage (layer 7) in Alexnet-face seems to give the best fit (or the second best fit) of the anterior face processing region AM across all datasets that they looked at. They then observe that this is not the case for ML/MF: sometimes it is best fit by layer 1 and sometimes it is better fit by higher layers (4 or 5); because there is more variability in terms of the Alexnet layer that best fits ML/MF from dataset to dataset, the authors conclude that it ML/MF is not as well explained by intermediate layers of convnets.

However, it seems that we could conclude something different -- indeed, we could conclude the

opposite of what the authors conclude -- if we focused on the VGG-Face network results rather than AlexNet. Looking at Fig. 9: (1) Across the board, in the VGG-face network, there is considerable consistency in terms of the layers (layer 4 and 5) that fit ML/MF best, which is comparable to what they report in Alexnet-face layers 6 and 7 vs. AM comparisons; and (2) As Fig. 9a shows, the VGG-Face network layer that fits both ML/MF and AM best (layer 5) correlates with ML/MF to higher degree relative to AM. This finding is in fact consistent with a report comparing VGG-face activations and ECoG recordings in humans (Grossman et al., 2018). Authors of this human neurophysiological study indeed conclude the opposite of the present paper: that intermediate stages of CNNs correspond well to how the brain processes faces, while later stages of CNNs do something different from what the brain does.

So, at a minimum, the story is more complicated than the paper suggests in its conclusion. Further complicating the interpretation is this: Despite the fact that there are higher correlations between layer 5 in the VGG-Face network and ML/MF, as the authors richly demonstrate, all of those intermediate and final stages across all networks lack the kind of shape tuning that is reported in the literature (Chang & Tsao; Ohayon et al.).

We believe that the above point arose because our initial presentation of the results on VGG-Face was not appropriate. We initially chose arbitrary layers in VGG-Face and overlaid the results on those from other Alexnet-based models, despite that VGG-Face has a deeper and different architecture. In the revision, we more rigorously chose the layer of VGG-Face that had the closest receptive field size as each layer of Alexnet architecture (Table S2). (Note that many early layers were skipped in VGG-Face since the growth of receptive field sizes in VGG-Face was much slower than AlexNet.) The revised plots are shown in Figure 9 in the new manuscript (and also below in this document).

However, the new plots (Figure 9) actually show that the results on VGG-Face are very similar to Alexnet-Face. Higher layers (Layers 6-7) give the best match with all of the tuning properties of AM, as in panels (a), (b), (c), and (d) (although the match in (b) is weak since VGG-Face can give only weak size-invariance; see (1-7) below). On the other hand, there is no layer that give simultaneously good matches with all of the tuning properties of ML. That is, the best matching layers are intermediate layers (Layers 4 to 5) for panel (a), higher layers (Layers 6-7) for panel (b), a lower layer (Layer 2) for panel (c), lower layers (Layers 1-3) for panel (e) red, lower-to-intermediate layers (Layers 1-5) for panel (e) blue, and no layer for panel (f). Since the newly chosen layers in the revised Figure 9 do not actually include relu5_3, which was Layer 5 in the old Figure 9, we show, for fairness, the results for all layers of VGG-Face in Figure R1 given in the end of this document. However, this did not change our conclusion, either. In particular, although relu5_3 gives a good match with ML for panels

(a), (b), and (e) blue, it gives rather a bad match with ML for panels (c) and (e) red.

Thus, in the revised manuscript, we maintain our original position that higher layers correspond to AM, but no layer corresponds to ML, across the different models that we have tested. However, we do think that our way of writing the conclusion in the initial manuscript was sometimes misleading as if we claimed that CNN higher layers can explain "fully" the properties of AM in (Freiwald and Tsao 2010) or explain any properties of AM beyond the four experiments considered, which is not our claim by any means. Thus, in the revision, we more carefully rephrased such statement to avoid such confusion throughout the manuscript.

Figure 9 (excerpt from the revised manuscript).

(1-2)

Overall, we believe that a more fair conclusion would be to say that CNNs trained to recognize faces do show some similarities to the brain's mechanisms but also a number of pronounced differences. They give rise to view-invariant coding at their higher layers as in the anterior regions in the brain, but both the nature of that code and the way it is computed from images might be quite different from the brain. None of the existing, standard CNN models gives a good account of the entire face patch system in the brain.

We believe that the conclusion suggested above is indeed consistent with our claim in our manuscript that higher layers give a good match with the multiple experimental data on AM, but no layer gives a good match with simultaneously those on ML. However, as mentioned in (1-1), our initial writing was sometimes misleading as if we claimed a much more general statement; we carefully rephrased such statement throughout the revised manuscript.

This point also hinted that our emphasis on considering multiple experiments was not strong enough in the initial manuscript, even though this was the most distinguishing point in our study. In the revision, we gave more stress on this point in Introduction and Discussion.

(1-3)

2. Relation to literature

When the authors compare their work to our earlier work, their characterization seems to us not entirely fair. They write: “Some previous studies have addressed a similar question but in a smaller scale. In one study (Yildirim et al., 2018), the authors tested view-identity tuning on a CNN model in comparison to experimental data (Freiwald and Tsao, 2010) and gave a similar result to ours (Figure 2). In another study (Chang and Tsao, 2017), the authors trained a smallish face-classifying CNN model and tested shape appearance tuning.”

They then describe some differences between their work and these earlier studies, and conclude: “Thus, the comparisons undertaken in our study here are far more thorough and extensive than those previous studies.”

We believe it's not fair to say simply that the present study is “far more thorough and extensive” than either our previous work or Chang and Tsao's. Those previous studies looked at fewer datasets, but analyzed them more deeply -- and arguably, much more “thoroughly”. Both our paper and Tsao's also presented additional, new data which the present paper does not consider. So the current work is importantly complementary to previous comparisons of CNNs and the macaque face patches, not strictly better. Its strength is to look more broadly at a wider range of previously published studies, which is valuable in a different and complementary way than the value of the earlier papers cited.

Indeed, our characterization of previous comparisons between CNN and face patches was not sufficient in mentioning compatible and complementary contributions in these two studies. Thus, in the revision, we additionally mention the experimental or theoretical context where the previous comparisons have been done and also mention the complementary contributions that were not considered in our study. We also modified the concluding remark to be more specific and avoided the word “thorough”.

However, regarding (Chang and Tsao 2017), we believe that our initial characterization was quite accurate. We indeed implemented and ran all analysis described in their paper and presented all results on multiple layers in our manuscript, while (Chang and Tsao 2017) presented results only partially, for a particular layer and a particular subset of the analysis. We indeed carefully looked again at their paper and confirmed that there was no deeper analysis or new data (in terms of CNN) that we did not consider. (Note that our initial manuscript had more data in Supplementary Materials that are not included in the main manuscript.) Thus, in the revision, we retain our initial position on the relationship with (Chang and Tsao 2017).

(1-4)

Another issue related to our work is how the authors characterize the new model we present, the “Efficient Inverse Graphics” or EIG network, and how it fits with their own interpretation of their findings. They argue -- at least, this is the part of their story we think is strongest (see earlier points of our review) -- that conventional CNNs capture some aspects of how the brain represents identity (or appearance?) in a viewpoint-invariant way, but not all aspects, and not how it gets there -- not the computations that lead to those viewpoint-invariant identity/appearance representations; also, conventional CNNs do not capture how the middle patches represent shape. That in a sense is also one of the core arguments in our paper; moreover, we present an alternative model -- an alternative kind of deep convolutional network, with a different training target, that is constructed to invert a graphics-like generative model, and which does capture these properties of the face patches (at least as represented in the Friewald and Tsao 2010 data) much better than standard CNNs. We have not yet tested our model on all the datasets that the present paper considers, which would be an important target for future work. The present paper inspires us to do that. But we believe it is important for the authors to acknowledge in a revised version of this paper that (1) the core argument it can sustain is also consistent with the findings and the argument we have made previously, looking at only a subset of the data considered here; and also (2) that we have proposed at least one model which takes the form of a deep convolutional network and which could potentially explain the full face-patch circuitry better than standard CNNs from the computer vision literature, but which needs to be further tested.

Although our initial manuscript had some comments corresponding to the points (1) and (2) above (in two separate paragraphs), we admit that these were too short. Thus, in the revised manuscript (Discussion section), responding to point (1) above, we mention that (Yildirim et al. 2018) has data and conclusion that are compatible with ours. Regarding point (2), we augment the part describing their alternative model to include more details on their model and mention their result giving a better

fit with monkey data than CNN.

The above comments also let us notice that our initial descriptions were too brief for other existing model studies, such as (Hosoya & Hyvärinen 2017). In the revision, we expand these parts with more details.

(1-5)

3. More minor comments.

It sounds strange and unscientific to use the word “smallish” to characterize the dataset in the Chang and Tsao paper. There must be a better word to use here?

In the revision, we remove this inessential emphasis.

(1-6)

The authors don't comment much on AL, and its mirror symmetry properties which have been the subject of a number of other modeling efforts (e.g., by Leibo and colleagues). For example, in Fig. 2, they see some mirror-symmetry in an earlier region which goes away or weakens in higher stages, which runs against at least a simple conclusion that only higher stages match anterior regions.

In our initial manuscript, we tended to refer to AM by "anterior", though it should also include AL in principle. In the revision, we try to make it more precise, using "AM", although we prefer still using "anterior" in the title and abstract for accessibility for the non-expert readers unfamiliar with this research area.

Besides, we have few comments on AL mainly because experimental data on AL is available in only one of the four experiments considered (Freiwald and Tsao, 2010) despite that our emphasis is to consider multiple experiments. Moreover, the mirror-symmetric tuning is replicated only partially in our CNN models (only profile views, but not half-profile views). Since we have only a weak result regarding to layer-to-patch correspondence for AL, we do not have much to claim in this respect. However, in the revision, we expand the parts in Discussion section that mention other studies replicating the mirror symmetry.

(1-7)

The size-invariance analysis is interesting and deserves more emphasis and discussion. Why is there such a big discrepancy across the board -- ML/MF, AL and AM are all more size-invariant than the CNNs, even more so than their highest stages? What would it take for a computational model to

match this constant and high-level of size invariance reported in the data?

The degree of size invariance obtained in the highest layer depends on how much variation in size is in the training data. In AlexNet-Face, we explicitly perform data augmentation up to four-times downsizing, which precisely corresponds to the size invariance index (1/4) in layer 7 in Figure 8b. However, we could downsize images further to, e.g., eight-times, which would give size invariance comparable to AM in principle. However, we could not since that would make the images too small (e.g., 28x28 relative to full size 224x224) and impossible to discriminate, which would badly affect the classification performance of the network. VGG-Face has a weak size invariance ($> 1/2$) since it was not trained with such data augmentation. In the revision, we add more on these points in Results section (Model variation).

(1-8)

In supplementary Fig S3, can the authors comment on why decoding accuracy decreases from layer 5 to layer 7? We found that surprising.

It was surprising to us, too. We can even observe such decrease from layer 3 to layer 5 and also the same tendency in Fig S1B. It is of course hard to understand precisely what happens in the intermediate layers of a deep net. Our tentative explanation is that lower layers have weaker nonlinearity and therefore are easier to decode by a linear regression. This point also implies that each single experiment has limitation in discriminating between plausible and implausible models, which emphasizes the importance of looking at multiple experiments. In the revision, we expand discussion on this point in Discussion section.

(1-9)

We were confused by the use of the word "explains": It is not clear what it means to say that CNNs explain some part of the face patch system according to the paper. If the authors want to use that word, then they should clarify the nature of that explanation. Is it just about matching or fitting the data, or is there some deeper sense in which the authors think their analyses explain how the neural circuits are working? Alternatively, the paper could refer to "match" or "fit" between CNNs and data, which is the more concrete result that they demonstrate.

In the present study, we mean by "explain" simply that the model gives a good match with the experimental tuning properties. In the literature, this usage of "explain" is also quite common (examples can be found in References in our manuscript). So, in the revision, we prefer using it in informal contexts like Introduction and Discussion, while we use "match" or "replicate" in a more

formal context such as in Results section.

Reviewer #2:

(2-1)

1. At first, it is not clearly described how the results from “the past experiment” in Fig. 1 are acquired. Did you (the authors) receive the results from the four different research groups? Did you receive the original neural responses? or just copy the values from the published papers? How can you sure that the neural responses from the past experiments and node activations from CNN are treated in the same way to draw the figures? Can we directly compare the neural responses and node activations without any normalization or fitting process? Explanations or references are required.

We did not use raw experimental data, but ran the same experimental protocols on the models (stimulus set and data analysis) and compared the population-level results with the corresponding published experimental data. We do not need normalization since data analysis does not depend on the response magnitude, nor fitting process since we compare the population-level results. We note this point in the revised manuscript (Results).

(2-2)

2. The authors used “mean STA correlation” from shape-appearance model (Chang and Tsao, 2017) to compare the view tolerance. This measure is the specific result from the shape-appearance model, but not direct measure to evaluate the view tolerance. In my opinion, the direct way to quantify the view tolerance would be to examine identity and view tuning curves, and compare the ratio between the variance across identities and the variance across views. I recommend to focus on discovering the actual difference between the ventral visual pathway and CNN, rather than reproducing the results from the previous works.

Since (Freiwald and Tsao 2010) provides a more direct way to measure identity selectivity and view invariance (their Fig 4GH), we additionally analyzed our mode layers using their method and compared with the experimental data (Figure S5 in the revised manuscript). The result shows pronounced differences from AM, with much weaker degree of view invariance to full profile (though reasonably strong for half profile). However, this is not surprising at all and in fact consistent with other parts of our results in Figure 2 and Figure 5. We add a note on this point in Results in the revision.

(2-3)

3. Overall, this manuscript is well written, and the results look clear. However, the contents related

to the size invariance and Fig. 3 (especially, Fig. 3B) are not easy to understand. To my understanding, the size invariance should be revealed as the “flat” tuning to the stimuli with different sizes. However, in Fig. 3B, the shapes of tuning curves look similar regardless of different layers. The only difference that I can find is the smaller average responses in higher layers. Why the smaller responses mean the higher invariance? In addition, the sentence “We then calculated, for each layer, the average responses, ... for each image size” is hard to read.

We quantify the degree of size invariance by how much robustly the population-level selectivity to faces over objects retains over different sizes. Note that, since this is the definition used in (Freiwald and Tsao 2010), we cannot change this for the purpose of comparison with the experimental results. Figure 3 shows that higher layers retain such selectivity for a wide range of image sizes, while lower layers do not. In the revision, we clarify this point (Results section). (Note that the average response magnitude is irrelevant here and throughout the paper (see (2-1) above). In CNN models we looked at, higher layers generally tended to have lower average magnitudes. However, all analyses have no dependency on average magnitudes.)

In the revision, we also rephrase the sentence "We then ..." pointed out above.

Figure R1: the summary plot for all the layers of VGG-Face network. The layers with shaded names correspond to those plotted in Figure 9.

Reviewers' comments:

Reviewer #2 (Remarks to the Author):

The authors have clearly addressed the comments, and I recommend this manuscript to be published in Communications Biology.

Reviewer #3 (Remarks to the Author):

In this paper, the authors investigate various deep neural network architectures, trained on large-scale face datasets, for their ability to mirror findings across the macaque face patch system. I appreciate the dedicated approach by the authors, who try and compare networks across a wide array of experimental findings to draw more generalisable conclusions. This important work is done too rarely with groups often focusing (and fitting) their models on individual datasets. The paper is accessible and overall written well (note that there are minor grammatical errors throughout, such as dropped definite/indefinite articles).

I am well aware that I am entering this process at a later stage than the other reviewers and will therefore try to not open too many new 'cans of worms'. Nevertheless, I have a few suggestions that I hope will help the authors strengthen their work.

Signed
Tim Kietzmann

Major:

1. The main claim of the paper (good agreement between model and brain for AM but not ML) is derived from the observation that consistent CNN layers are selected for AM, whereas there is larger variability for ML. While the authors focus on multiple experiments that all investigate the same target area, substantial differences in the experimental setups will exist across studies (degrees visual angle, stimulus materials, presentation timing, different alignment in electrode placement, etc.). At the same time, the tested model architectures are neither fitted to the data nor are the model RF properties matched against the biological system. This raises the question in how far a perfect agreement for layer-selection across experiments is to be expected in the first place. As a simple example, consider two studies in which stimuli are shown with different sizes. This could lead to a different bias in the neural data and therefore to a differential selection of model layers (mirroring different model RF sizes for example). Based on this reasoning, I think that the requirement of the exact same network layer is too stringent. As a solution, the authors could compare the relative impact of earlier vs later model layers. Perhaps contrary to the current conclusion in the manuscript, this approach reveals that this general trend holds true for many of the analyses: ML is better predicted by earlier layers, whether later layers better mirror selectivity in AM. The consistent final layer selection for AM is striking, but the disagreement for ML is not a deal-breaker either. The latter does, however, point towards a more complicated story, the origins of which need to be determined in future work (architecture design, experimental differences, or general disagreement). I therefore invite the authors to acknowledge and discuss this possibility in the paper, and to adjust their conclusions accordingly.

2. A lot of the results presented rely on "statistics by eye". The authors describe qualitatively whether some layers explain the data better than others, but the quantitative backing with statistical analyses is missing. This invites unwanted subjectivity into the analyses. Are any of the differences observed

significant? For the most part, we do not know. As an illustrative example, I find the fit in Figure 4B, layer 1, between model and ML quite remarkable, but the authors describe it as “not particularly similar”, and later summarise as “lacking [of] ML-like properties”. To me, this is too strong a conclusion, especially in light of missing statistical tests. I am aware that statistical tests are complicated by the fact that the authors only trained a single network instance. As a way forward, it would therefore be great to consider alternative models (e.g. untrained baseline models, etc), so that we know how bad the trained CNNs really are. It could well be that they are better than most if not all other image-computable models that we have.

Minor:

3. Can the authors comment on the fact that the size invariance estimates of the brain data are exactly identical across face patches? This seems unlikely to me.

4. I think this paper can stand on its own without downplaying previous work to enhance the feeling of novelty. For example, I feel that the work and results by Yildirim et al. are downplayed too much in the introduction. The sentence (l. 42-47) seems is ambiguous at best, hiding the fact that they did use different models, multiple face patches, and I would not call it a “side question” of their work.

5. Can the authors comment further on why the position invariance results could not be tested? Would this not be possible with the smaller face stimuli used for the size-invariance tests?

6. l 387: The crucial importance of recurrent connectivity in CNNs has been investigated in depth in Kar et al. (2019) and Kietzmann et al. (2019). Disclaimer: the latter is obviously a plug for our own work.

7. Is Figure 1 showing a human brain, not a macaque?

Revision of manuscript "CNN explains tuning properties of anterior, but not middle, face-processing areas in macaque IT"

We would like to thank the reviewers for very useful comments. We have made a major revision of our previously submitted manuscript.

Reviewer #3 made major comments on potential confound of the compared experimental data and missing statistical analysis for backing our claims, as well as minor comments regarding more specific analysis and points to discuss. We made the best efforts to deal with all these, which resulted in a major recalculation of the entire analysis and a major rewriting and restructuring of the entire manuscript. The specific modifications are indicated by blue fonts in the re-submitted manuscript.

Below are our point-by-point replies to the comments. Each individual comment was given numbers in red font and our response and reaction to the comment was described in black font.

Reviewer #3:

1. The main claim of the paper (good agreement between model and brain for AM but not ML) is derived from the observation that consistent CNN layers are selected for AM, whereas there is larger variability for ML. While the authors focus on multiple experiments that all investigate the same target area, substantial differences in the experimental setups will exist across studies (degrees visual angle, stimulus materials, presentation timing, different alignment in electrode placement, etc.). At the same time, the tested model architectures are neither fitted to the data nor are the model RF properties matched against the biological system. This raises the question in how far a perfect agreement for layer-selection across experiments is to be expected in the first place. As a simple example, consider two studies in which stimuli are shown with different sizes. This could lead to a different bias in the neural data and therefore to a differential selection of model layers (mirroring different model RF sizes for example). Based on this reasoning, I think that the requirement of the exact same network layer is too stringent. As a solution, the authors could compare the relative impact of earlier vs later model layers. Perhaps contrary to the current conclusion in the manuscript, this approach reveals that this general trend holds true for many of the analyses: ML is better predicted by earlier layers, whether later layers better mirror selectivity in AM. The consistent final layer selection for AM is striking, but the disagreement for ML is not a deal-breaker either. The latter does, however, point towards a more complicated story, the origins of which need to be determined in future work (architecture design, experimental differences, or general disagreement). I therefore invite the authors to acknowledge and discuss this possibility in the paper, and to adjust their conclusions accordingly.

Indeed, we crucially assume that the investigated populations in the previous experimental studies represent the entire population of the target face patch (by cell sampling without much bias). However, there is no information supporting or contradicting this assumption in the original experimental studies. Thus, we cannot completely exclude the possibility that the four past monkey experiments could have hit different clusters of neurons in each face patch, so that their reported quantitative results could have been on different clusters. Thus, despite that we could not find a precise layer-to-patch correspondence, it still remains possible that some other more complicated correspondence may exist. In the revision, we acknowledge this possibility and weaken our conclusion accordingly.

Although the reviewer mentions other potential factors, we believe these are much less crucial. First, the stimulus sizes are actually similar across the four experiments, ranging from 5.7 to 7 degrees. Second, the presentation durations are different but also not so drastically different, 117 ms to 200 ms per stimulus, although 200ms intervals were taken in two studies but not in two others. Third, the number of repetitions for each stimulus image is similar: 3-5 times in three studies or 3-10 times in one study.

The reviewer suggests to "compare the relative impact of earlier vs later model layers", which is in fact more or less what we do in the section "Summary of Correspondence". With this analysis, the result indicates that different layers give the best match for results of different experiments on ML (middle layers for Figure 8a; higher layers for Figure 8b; lower layers for Figure c, e). Thus, our approach is more or less in line with the review's viewpoint, where the essential difference lies in the expectation of a layer simultaneously reproducing the multiple experimental data, which is discussed above.

2. A lot of the results presented rely on "statistics by eye". The authors describe qualitatively whether some layers explain the data better than others, but the quantitative backing with statistical analyses is missing. This invites unwanted subjectivity into the analyses. Are any of the differences observed significant? For the most part, we do not know. As an illustrative example, I find the fit in Figure 4B, layer 1, between model and ML quite remarkable, but the authors describe it as "not particularly similar", and later summarise as "lacking [of] ML-like properties". To me, this is too strong a conclusion, especially in light of missing statistical tests. I am aware that statistical tests are complicated by the fact that the authors only trained a single network instance. As a way forward, it would therefore be great to consider alternative models (e.g. untrained baseline models, etc), so that we know how bad the trained CNNs really are. It could well be that they are better than most if not

all other image-computable models that we have.

First, we had quantitative evaluation in the section "Summary of Correspondence" in the initial manuscript, but did not use in the beginning of Result section showing individual data. Therefore, in the revision, we quote some of those numbers in the earlier section to back up our comparison.

In addition, we introduce the following statistical tests for further back-up (describing the details in Methods section):

- For view-identity tuning, we test whether each RSM correlation lies outside the 2SD range of the random cases, i.e., correlation coefficients between each experimental RSM and a (repeatedly generated) random RSM and then (Figure 8a). (That is, we test if the same level of correlation may arise from random RSMs with probability less than 5%). In the result, the random cases generally gave very small correlations. Note that, since we do not have raw neural response data, it is not possible to perform bootstrapping; this prevents us to test how much closer a model layer is to different face patches. (The authors of the experiment gave us the experimental RSM but not the raw data; the figures in their article did not have enough resolution to extract such data).
- For shape-appearance preference, we calculate the 95% confidence intervals of mean SPIs for ML, AM, and their midpoint (Figure 8c) by bootstrapping on the experimental data. This time was lucky since we were able to reconstruct the SPI values for individual neurons from the scatter plot (Figure 1E) in (Chang & Tsao 2017). As a result of this analysis, our initial claim regarding Figure 4B, layer 1, turned out indeed too strong as the reviewer pointed out (layer 1 actually significantly deviates from the midpoint of ML and AM). Therefore, in the revision, we change it so that layer 1 gives "an intermediate result between AM and ML, somewhat closer to ML". However, even with this result, we believe that the similarity with ML in layer 1 is not as strong as the similarity with AM in layer 7. (This view is also supported by the result of the next analysis on facial feature decoding from the same experiment, described below.) Therefore, in the revision, we weaken our summary statement, but in a moderate way, that "in terms of shape-appearance tuning, the CNN model exhibited prominently AM-like properties but less clearly ML-like properties".
- For facial feature decoding, we calculate the root mean squared error between the explained variances between model and experiment (AM or ML) and show 2SD ranges after 100 times re-sampling of model subpopulation. This shows a good match between higher layers and AM but poorer matches between lower-to-intermediate layers and both AM and ML. Since our manuscript is running short of space, we move this result (less crucial than other results) to Supplementary Figure 4.

- For view tolerance, we test whether each mean correlation lies outside the 2SD range of the random cases, i.e., mean correlations between random STA vectors (Figure 8d). The random cases generally gave very small mean correlations.
- For facial geometry and contrast polarity tuning, we test whether each cosine similarity lies outside the 2SD range of random cases, i.e., cosine similarities between each experimental distribution and random distributions (Figure 8e,f). In the result, the 2SD ranges are generally quite large and, as a result, only lower layers show significance for feature-per-unit and unit-per-feature. For contrast polarity, no layer shows notable significance.
- For size invariance, we could not find any meaningful statistical method to measure significance since our definition of size invariance index involves a coarse quantization, which prevents any appreciable variability. (Note that the size invariance index can take 1 at maximum, which could serve as a baseline built in the definition.) See also point 3 below.

The reviewer suggests using an untrained model as baseline. However, we are not sure if this serves as a good baseline since it has a particular computational process with multiple stages and local feature representations even though the weights are random. In fact, an untrained model showed large discrepancy for some tuning properties, but surprising similarity for other properties. Therefore, in the revision, we add these results to Figure 9 as an additional "CNN model instance" and add some points to discuss.

We also tried bootstrapping on the model results by re-sampling the model population. However, we did not find it meaningful since this gave only very narrow confidence intervals that were completely invisible in the figures and almost always gave significance to even extremely small differences.

3. Can the authors comment on the fact that the size invariance estimates of the brain data are exactly identical across face patches? This seems unlikely to me.

The measure we used gives the same quantity for all face patches from Figure S10C of Freiwald and Tsao (2010).

However, in the revised Supplementary Information, we add the second analysis (Supplementary Figure 2) on size invariance using a different measure based on Figure S10D of the same experimental study, which gives different quantities for ML and for AL/AM. The result is consistent with our first analysis on size invariance. (We tried also statistical test similarly to point 2 above but the 2SD range was concentrated to nearly zero.) We prefer to keep using the first analysis in the main text (instead

of the second one) since the criterion seems more direct for measuring invariance.

4. I think this paper can stand on its own without downplaying previous work to enhance the feeling of novelty. For example, I feel that the work and results by Yildirim et al. are downplayed too much in the introduction. The sentence (l. 42-47) seems is ambiguous at best, hiding the fact that they did use different models, multiple face patches, and I would not call it a “side question” of their work.

In the revision, we modify the mention on the previous CNN comparisons in the introduction and discussion so that the downplaying impression is weakened.

5. Can the authors comment further on why the position invariance results could not be tested? Would this not be possible with the smaller face stimuli used for the size-invariance tests?

In Freiwald and Tsao (2010), they used the same size of face images for the main experiment and for the position-invariance experiment. If we followed the same convention, then the face image would exceed the boundary and give a meaningless result, which is why we did not include this test in the initial submission.

However, in the revised Supplementary Information, we add a modified version of the position invariance test using half-size stimuli with a lesser extent of position variation (Supplementary Figure 2). Note that this is not meant to be a fair comparison, but only for reference, though the result is reasonably consistent with the experiment.

6. I 387: The crucial importance of recurrent connectivity in CNNs has been investigated in depth in Kar et al. (2019) and Kietzmann et al. (2019). Disclaimer: the latter is obviously a plug for our own work.

We add both in the revision.

7. Is Figure 1 showing a human brain, not a macaque?

We replace it with a cartoon macaque brain.

REVIEWERS' COMMENTS:

Reviewer #3 (Remarks to the Author):

The authors have sufficiently addressed my previous comments.